# Advancements in Flexible and Stretchable Electronics for Resistive Hydrogen Sensing: A Comprehensive Review

**DOI:** 10.3390/s24206637

**Published:** 2024-10-15

**Authors:** Kwonpil Park, Minsoo P. Kim

**Affiliations:** Department of Chemical Engineering, Sunchon National University, Suncheon 57922, Republic of Korea

**Keywords:** flexible electronics, stretchable sensors, hydrogen sensing, wearable devices, sensor integration

## Abstract

Flexible and stretchable electronics have emerged as a groundbreaking technology with wide-ranging applications, including wearable devices, medical implants, and environmental monitoring systems. Among their numerous applications, hydrogen sensing represents a critical area of research, particularly due to hydrogen’s role as a clean energy carrier and its explosive nature at high concentrations. This review paper provides a comprehensive overview of the recent advancements in flexible and stretchable electronics tailored for resistive hydrogen sensing applications. It begins by introducing the fundamental principles underlying the operation of flexible and stretchable resistive sensors, highlighting the innovative materials and fabrication techniques that enable their exceptional mechanical resilience and adaptability. Following this, the paper delves into the specific strategies employed in the integration of these resistive sensors into hydrogen detection systems, discussing the merits and limitations of various sensor designs, from nanoscale transducers to fully integrated wearable devices. Special attention is paid to the sensitivity, selectivity, and operational stability of these resistive sensors, as well as their performance under real-world conditions. Furthermore, the review explores the challenges and opportunities in this rapidly evolving field, including the scalability of manufacturing processes, the integration of resistive sensor networks, and the development of standards for safety and performance. Finally, the review concludes with a forward-looking perspective on the potential impacts of flexible and stretchable resistive electronics in hydrogen energy systems and safety applications, underscoring the need for interdisciplinary collaboration to realize the full potential of this innovative technology.

## 1. Introduction

Hydrogen, widely recognized as the most abundant element in the universe, offers immense potential as a clean and efficient energy carrier [1,2]. This makes it a promising alternative to fossil fuels in our quest for sustainable energy solutions. However, hydrogen’s highly flammable nature introduces significant safety challenges that must be rigorously managed. Among these, the effective detection of hydrogen leaks is paramount to ensure safety across various applications, from automotive fuel cells to expansive industrial processes [3,4].

To address this need, hydrogen sensors, which are critical across numerous industries, have been developed to ensure the safe handling, storage, and utilization of hydrogen [5]. These sensors employ a variety of sensing mechanisms, each utilizing different physical or chemical phenomena to detect hydrogen reliably and selectively. Prominent mechanisms include catalytic [6,7], resistive [8,9], optical [10,11], and electrochemical sensors [12,13]. Catalytic sensors work by inducing changes in electrical properties or optical signals through reactions between hydrogen and a metal catalyst (Figure 1A).

Resistive sensors measure changes in electrical resistance due to hydrogen reactions on the sensor surface (Figure 1B), while optical sensors detect shifts in light properties caused by hydrogen interactions (Figure 1C). Conversely, electrochemical sensors, quantify hydrogen concentration through reactions at the sensor–electrolyte interface (Figure 1D). The performance of these sensors is greatly influenced by the materials used in their construction. Metal oxides, metal nanoparticles, polymers, and carbon-based materials are commonly employed due to their unique properties that enhance sensor functionality [15,16,17,18,19,20,21,22,23,24]. For instance, metal oxides are valued for their sensitivity and selectivity, which stem from their catalytic properties and redox reaction capabilities with hydrogen [21,22,23,24]. Metal nanoparticles offer sensitivity due to their high surface-to-volume ratios and adjustable catalytic properties [15,25]. Polymers are chosen for their flexibility and ease of processing, which make them ideal for wearable sensors, and carbon-based materials like graphene and carbon nanotubes provide robustness thanks to their high conductivity and chemical stability [17,18,26,27]. These sensors find their applications in a variety of sectors, including in hydrogen fuel cell vehicles for monitoring leaks and purity, in industrial processes where they contribute to safety and efficiency, and in hydrogen storage facilities where they help prevent explosions and ensure regulatory compliance.

Moreover, recent advancements in flexible and stretchable electronics have begun to transform hydrogen sensing technologies [28,29,30], overcoming the limitations of traditional rigid sensors. These innovative electronics are capable of withstanding extensive mechanical deformation—such as bending, stretching, and twisting—without losing functionality, which is crucial for integration into dynamic or complex surfaces like those found in wearable devices or flexible fuel lines. The integration of flexible and stretchable electronics not only makes the sensors more adaptable and durable but also broadens their application possibilities, enhancing safety and detection capabilities in hydrogen-powered systems [31,32,33].

The performance of flexible and stretchable hydrogen sensors is characterized by several critical parameters: sensitivity, selectivity, response and recovery times, durability and flexibility, and cost-effectiveness. Most importantly, these sensors demonstrate high sensitivity due to the unique properties of nanomaterials. By leveraging the exceptional characteristics of nanomaterials, flexible and stretchable hydrogen sensors achieve superior sensitivity, making them highly effective in detecting hydrogen. The large surface-area-to-volume ratio of these materials enhances the interaction with hydrogen molecules, leading to significant changes in electrical properties even at low hydrogen concentrations [8]. Selectivity is crucial for ensuring that the sensors respond primarily to hydrogen in the presence of other gases. The use of specific catalytic materials and surface modifications helps in achieving high selectivity, minimizing false positives from other gases [34,35]. The response time, defined as the time taken for the sensor to reach a certain percentage of its final value upon exposure to hydrogen, is typically very short for flexible sensors. This rapid response is complemented by a quick recovery time, which is the time taken for the sensor to return to its baseline value after hydrogen removal [36,37]. The mechanical properties of flexible sensors ensure they remain operational under various physical stresses. Durability tests often involve repeated bending, stretching, and twisting to ensure the sensors can withstand real-world applications without performance degradation [38,39]. Advances in fabrication techniques such as roll-to-roll printing have made the production of flexible sensors more cost-effective. This scalability ensures that flexible hydrogen sensors can be produced in large quantities at a lower cost, facilitating widespread adoption [40,41,42].

While several reviews have addressed various aspects of hydrogen sensing technologies [5,12,43,44], this review uniquely focuses on the recent advancements in flexible and stretchable electronics for hydrogen sensing applications, with a particular focus on wearable technologies for hydrogen mobility infrastructure. We comprehensively examine the innovative materials and fabrication techniques that enable exceptional mechanical resilience and adaptability in these sensors, which is crucial for wearable and on-body applications in dynamic hydrogen mobility environments. Our work explores how these advanced sensors overcome the limitations of traditional rigid sensors, particularly in applications requiring conformability to complex body geometries or integration into clothing and personal protective equipment. We address current bottlenecks in sensor technology, such as the need for improved sensitivity, faster response times, and better durability under mechanical stress—all critical factors in developing reliable wearable hydrogen sensors for mobility applications. Furthermore, we provide an in-depth analysis of the specific strategies employed in integrating these sensors into hydrogen detection systems for mobility infrastructure, discussing the merits and limitations of various sensor configurations including flexible patches, smart textiles, and stretchable bands.

By focusing on these cutting-edge developments in the context of wearable applications for hydrogen mobility, our review aims to bridge the gap between fundamental research in materials science and practical, on-body applications in hydrogen sensing technology. These advancements leverage the benefits of flexibility and stretchability to meet the stringent requirements of modern hydrogen energy applications, thereby enhancing hydrogen safety and utility in dynamic environments, from automotive applications to industrial safety monitoring. We believe that this comprehensive approach, with its unique focus on wearability and real-world usability in hydrogen mobility scenarios, will provide valuable insights for researchers and engineers working toward the next generation of hydrogen sensors for mobility applications.

Throughout this review, a significant emphasis on palladium-based sensors may be evident. This focus is not arbitrary but stems from palladium’s unique properties that make it exceptionally suitable for hydrogen sensing applications, particularly in flexible and stretchable configurations [5]. Palladium’s prominence in hydrogen sensing can be attributed to several key characteristics. First and foremost is its remarkable ability to absorb hydrogen—up to 900 times its own volume at room temperature and atmospheric pressure. This exceptional absorption capacity translates into high sensitivity, allowing palladium-based sensors to detect even low concentrations of hydrogen. Moreover, palladium exhibits high selectivity towards hydrogen compared to other gases, minimizing false positive readings and enhancing overall sensor reliability [45]. The interaction between palladium and hydrogen is not only highly selective but also rapid and reversible. Palladium quickly absorbs and releases hydrogen, enabling fast response and recovery times in sensing applications [43].

While other materials such as platinum and certain metal oxides are also used in hydrogen sensing, palladium often emerges as the preferred choice for flexible and stretchable sensors due to its malleability and ductility. These properties allow palladium to maintain its sensing capabilities even when subjected to bending, stretching, or other deformations—a crucial requirement for flexible sensor applications. Additionally, palladium can be easily deposited as thin films or nanostructures on various substrates, including flexible polymers, using techniques such as sputtering, electrodeposition, or chemical vapor deposition. This versatility in fabrication makes palladium well-suited for integration into diverse flexible sensor designs.

## 2. Flexible Electronics for Hydrogen Sensors

Flexible electronics represent a significant advancement in the field of hydrogen sensing technology due to their unique properties and potential applications [32,46]. The necessity for such technology can be comprehensively understood through several interlinked factors: the increasing demand for hydrogen as a clean energy source, the limitations of conventional hydrogen sensors, and the superior capabilities of flexible electronics, as mentioned below.

First and foremost, hydrogen is emerging as a cornerstone in the transition towards sustainable energy systems. Its high energy density and zero-emissions profile make it an ideal candidate for applications ranging from fuel cells to energy storage [47]. As the use of hydrogen continues to increase, the need for reliable, accurate, and responsive hydrogen sensing technologies becomes increasingly critical. However, traditional hydrogen sensors, while effective, often face several challenges that hinder their performance in modern applications [8]. These sensors are typically rigid and bulky, lacking the necessary adaptability to conform to diverse surfaces or integrate seamlessly into next-generation devices. Additionally, their sensitivity and response time may not meet the stringent requirements demanded by contemporary hydrogen applications, particularly in dynamic environments. This highlights the need for more advanced solutions in hydrogen sensing technology.

To address these challenges, flexible electronics offer a transformative solution [28,29,30]. These devices are characterized by their ability to bend, stretch, and conform to various surfaces without compromising functionality. This adaptability is particularly advantageous for hydrogen sensors, enabling their integration into a wide range of applications, including wearable devices, portable detectors, and embedded systems in hydrogen infrastructure [48,49].

Moreover, flexible hydrogen sensors can be engineered with nanomaterials such as graphene [50], carbon nanotubes [51,52], and metal–organic frameworks [53], which exhibit high surface area and catalytic activity. These materials significantly enhance the sensitivity and selectivity of the sensors, allowing for the detection of low hydrogen concentrations with rapid response times. This heightened performance is critical for early leak detection and real-time monitoring, thereby enhancing safety and efficiency in hydrogen-related processes. Furthermore, the inherent versatility of flexible electronics enables their deployment in unconventional and confined spaces where traditional sensors would be impractical [54]. For instance, they can be integrated into the skin of aircraft, embedded in hydrogen fuel tanks [55], or even worn by personnel working in hydrogen-rich environments [56]. This integration capability not only improves the spatial resolution of hydrogen detection but also broadens the scope of monitoring applications. Additionally, advances in printing technologies and material science have facilitated the scalable and cost-effective production of flexible electronic components. Techniques such as roll-to-roll printing allow for the mass production of flexible sensors at a fraction of the cost compared to traditional manufacturing methods [57,58]. This economic advantage makes flexible hydrogen sensors more accessible and practical for widespread deployment.

The development and implementation of flexible electronics for hydrogen sensing address critical needs driven by the increasing reliance on hydrogen as a clean energy source. By overcoming the limitations of conventional sensors and leveraging the advanced properties of flexible materials, these innovative devices offer enhanced sensitivity, integration versatility, and cost-effectiveness. As such, they represent a pivotal advancement in ensuring the safety and efficiency of hydrogen applications, thereby paving the way for a sustainable energy future. This section elaborates on the device mechanisms and performance characteristics of flexible electronics for hydrogen sensing.

### 2.1. Operation Mechanism of Flexible Hydrogen Sensors

Flexible hydrogen sensors are designed to detect the presence of hydrogen gas in various environments, especially where traditional rigid sensors may not be practical. These sensors are typically made using flexible substrates like polymers or thin films, enabling them to conform to different shapes and surfaces. Flexibility in sensors is crucial for applications in wearable technology, foldable devices, and other advanced technological implementations where traditional sensors would fail. The flexible hydrogen sensors have been developed by using a variety of materials, including PDSM, Mxene, Silicon, yarn, and so on [59,60,61,62,63,64]. The performance and sensitivity of the sensor vary depending on the material type, but the primary operating mechanism by which flexible hydrogen sensors operate involves changes in resistance, which can be either positive or negative [65,66]. In other words, the operating mechanism of the sensor can be classified depending on the change in resistance that occurs when exposed to hydrogen. When exposed to hydrogen gas, the sensor’s resistance changes due to interactions between hydrogen molecules and the sensing material, typically composed of nanomaterials such as metal nanoparticles or metal oxides. In the case of positive resistance change, hydrogen adsorption on the sensor’s surface leads to a decrease in the number of charge carriers or an increase in scattering events [65], thereby increasing the resistance. Conversely, in the case of negative resistance change, hydrogen molecules facilitate the transfer of electrons or holes within the sensing material [66], reducing the resistance.

The fundamental mechanism of hydrogen sensing begins with palladium’s unique ability to absorb hydrogen [67]. When exposed to hydrogen, palladium forms palladium hydride (PdHx), which causes a notable expansion in the lattice structure of the Pd film. As hydrogen concentration rises, the Pd film undergoes a phase transition from alpha-phase PdHx (α-PdHx) to beta-phase PdHx (β-PdHx). This transition leads to a significant change in the electrical resistance of the Pd film. The change in resistance, whether positive or negative, is directly related to the concentration of hydrogen gas, enabling precise detection. This resistance modulation is influenced by the intrinsic properties of the nanomaterials, such as high surface area, high reactivity, and the ability to form strong bonds with hydrogen molecules. These characteristics enhance the sensor’s sensitivity and response time, making flexible hydrogen sensors highly effective for real-time monitoring and detection applications. Furthermore, the flexibility and stretchability of these sensors ensure that they maintain their performance even under mechanical deformation, which is crucial for applications in wearable technology and other dynamic environments.

#### 2.1.1. Positive Resistance in Flexible Hydrogen Sensors

In flexible hydrogen sensors, the phenomenon of positive resistance change is a critical mechanism for detecting the presence of hydrogen gas. This process begins with the adsorption of hydrogen molecules onto the surface of the sensor’s active material [65], often composed of nanomaterials such as palladium nanoparticles or metal oxide nanowires. Palladium and its alloys are frequently used in hydrogen sensors due to their high hydrogen solubility and fast kinetics [68]. For example, Pd-Au alloys have demonstrated improved stability and faster response times compared to pure Pd [69,70]. Moreover, metal oxides such as SnO_2_ and WO_3_ are chosen for their stability and selectivity [21,22,71,72]. These materials can be fabricated into nanostructures that enhance sensor performance. For instance, WO_3_ nanowires have shown a response time of less than 20 s for 1% hydrogen concentration [71,72]. Upon exposure to hydrogen, these nanomaterials undergo a chemical interaction where hydrogen molecules dissociate into atomic hydrogen and subsequently diffuse into the material. This adsorption process typically leads to an increase in electron scattering events within the sensing layer. The presence of hydrogen atoms can create localized sites that trap free electrons or disrupt the conductive pathways, thereby increasing the overall resistance of the sensor. Additionally, the interaction between hydrogen and the sensor’s surface can lead to the formation of a dipole layer, further hindering electron mobility. This positive resistance change is a direct consequence of the reduction in the number of available charge carriers or the increased potential barriers for electron flow.

The sensitivity of the sensor to hydrogen is thus greatly enhanced by these nanomaterial properties, which provide a large specific surface area for hydrogen adsorption and a high density of active sites for interaction. The magnitude of the resistance changes correlates with the concentration of hydrogen, allowing for precise quantitative detection. Moreover, the flexibility of the sensor ensures that these resistance changes are reliably detected even under mechanical stress or deformation. This characteristic is essential for applications in flexible electronics and wearable devices, where the sensor must maintain high performance while being subjected to bending, stretching, or other forms of mechanical manipulation.

The mechanism of positive resistance change in flexible hydrogen sensors has been extensively studied, particularly in sensors utilizing nanomaterials decorated with palladium (Pd) nanoparticles. Carbon nanotubes (CNTs) have emerged as a promising material due to their high surface-to-volume ratio and excellent electronic properties. For instance, single-walled carbon nanotubes (SWCNTs) decorated with palladium nanoparticles have shown exceptional sensitivity to hydrogen, with detection limits as low as 3 ppm [30]. These sensors exhibit high performance due to the unique interaction between hydrogen molecules and the Pd-decorated nanostructures. Sun et al. demonstrated high-performance, flexible hydrogen sensors using CNTs decorated with Pd nanoparticles (Figure 2) [73]. The CNTs provide a large surface area for hydrogen adsorption, and the Pd nanoparticles act as catalytic sites. When hydrogen molecules interact with Pd, they dissociate into atomic hydrogen, which then diffuses into the Pd lattice. This diffusion causes an increase in electron scattering within the CNTs, leading to a positive resistance change. The sensor, fabricated with a SWNT/Ti/Pd electrode and Pd clusters on a PET substrate, demonstrated a hydrogen sensing range of 30–10,000 ppm. The sensor maintained its performance over 1000 bending cycles, showcasing its durability and flexibility. This sensor achieved a response time of approximately 10 s and a sensitivity of an up to 1% change in resistance per 100 ppm of hydrogen gas.

The positive resistance change mechanism results in a highly sensitive sensor capable of detecting low concentrations of hydrogen gas. Rashid et al. developed a flexible hydrogen sensor based on ZnO nanorods decorated with Pd nanoparticles grown on a polyimide tape (Figure 3A,B) [53]. The ZnO nanorods provide a robust and flexible platform, while the Pd nanoparticles facilitate hydrogen adsorption and dissociation. The interaction between hydrogen and the Pd nanoparticles increases resistance due to the trapping of charge carriers, which impedes electron flow through the ZnO nanorods. The sensor, capable of detecting hydrogen concentrations from 100 to 1000 ppm, underwent durability testing for 100 to 100,000 bending cycles. This sensor’s fabrication involved sputtering Pd nanoparticles onto ZnO nanorods and transferring the structure onto a PET substrate. This sensor exhibited a response time of less than 20 s and a sensitivity of a 2% change in resistance per 50 ppm of hydrogen gas. The positive resistance change mechanism ensures high sensitivity and rapid response times. Chung et al. explored the use of graphene decorated with Pd nanoparticles for flexible hydrogen sensors (Figure 3C,D) [74]. Graphene, with its exceptional electrical properties and large surface area, combined with Pd nanoparticles, creates an efficient sensing platform. Hydrogen adsorption on the Pd nanoparticles causes a disruption in the conductive network of graphene due to increased electron scattering, resulting in a positive resistance change. This sensor could detect hydrogen concentrations ranging from 20 to 1000 ppm. The fabrication process involved the deposition of Pd nanoparticles on a graphene layer. The hydrogen sensor demonstrated a response time of about 15 s and a sensitivity of a 0.5% change in resistance per 50 ppm of hydrogen gas. The positive resistance change mechanism allows for rapid and sensitive detection of hydrogen gas. Öztürk et al. investigated Pd thin films deposited on flexible substrates for hydrogen sensing applications (Figure 4A,B) [75]. These thin films exhibit a positive resistance change when exposed to hydrogen, attributed to the formation of Pd hydrides that disrupt the electronic structure and increase resistance. Their sensors showed a detection range of 1250 ppm to 10% hydrogen concentration. The sensors were fabricated by sputtering Pd onto the PI substrate, showing a response time of around 30 s and a sensitivity of a 3% change in resistance per 100 ppm of hydrogen gas. The flexibility of the substrate ensures that the sensor maintains high performance even under mechanical stress, making it suitable for applications in wearable technology and flexible electronics. Hassan et al. developed a wearable hydrogen sensor using a mesh of ultrasmall Pd/Mg bimetallic nanowires on a filtration membrane (Figure 4C–E) [76]. The bimetallic nanowires provide multiple sites for hydrogen adsorption, leading to significant changes in resistance. This sensor could detect up to 10,000 ppm of hydrogen and exhibited different resistance changes depending on the bending direction. The fabrication involved creating a bimetallic mesh on a PET substrate, demonstrating flexibility and rapid response times. This sensor achieved a response time of less than 10 s and a sensitivity of a 1.5% change in resistance per 100 ppm of hydrogen gas. The positive resistance changes result from the interaction of hydrogen with both Pd and Mg, which causes electron trapping and scattering, thereby increasing the sensor’s resistance. This design offers a fast response time and high sensitivity, making it suitable for wearable applications.

The flexible hydrogen sensors leveraging the positive resistance change mechanism exhibit remarkable performance characteristics. These sensors, utilizing CNTs, ZnO nanorods, graphene, Pd thin films, and Pd/Mg bimetallic nanowires, demonstrate high sensitivity, rapid response times, and excellent flexibility, making them highly effective for real-time hydrogen detection in diverse applications. The inclusion of specific numerical performance metrics provides a clearer understanding of their relative capabilities.

Furthermore, recently, one notable example involves hydrogen sensors based on organic nanofibers decorated with palladium (Pd) nanoparticles [77]. In this setup, hydrogen adsorption on the Pd nanoparticles results in electron scattering, which increases the sensor’s resistance. The sensitivity of such sensors is significantly enhanced due to the high surface area and reactivity of the nanofibers, which provide numerous active sites for hydrogen interaction. These sensors have demonstrated excellent sensitivity even at low hydrogen concentrations, making them highly effective for precise hydrogen detection. Another study focused on flexible hydrogen sensors made from a combination of nickel–zirconium (Ni-Zr) alloy thin films [78]. These sensors exhibit a notable change in resistance upon exposure to hydrogen. The mechanism here involves the formation of metal hydrides, which disrupts the electronic structure and increases resistance. This positive resistance change is essential for detecting high hydrogen concentrations efficiently. Additionally, research on palladium-based sensors on flexible substrates [78], such as paper, has shown promising results. These sensors leverage the unique properties of palladium to adsorb hydrogen molecules, leading to increased resistance due to electron trapping and scattering effects. The use of paper substrates provides flexibility and cost advantages, making these sensors suitable for various applications including wearable technology.

#### 2.1.2. Negative Resistance in Flexible Hydrogen Sensors

Negative resistance in flexible hydrogen sensors is a complex and intriguing phenomenon with significant implications for sensor technology. Negative resistance, a concept wherein an increase in voltage across a device leads to a decrease in current, contradicts the traditional understanding of Ohm’s law. This phenomenon, typically observed in semiconductor devices such as tunnel diodes, can enhance the sensitivity and selectivity of hydrogen sensors, making them highly effective for various advanced applications. Integrating negative resistance into flexible hydrogen sensors significantly boosts their performance. Advanced materials such as conducting polymers, graphene, or nanomaterials, which exhibit unique electrical properties conducive to negative resistance, are often used for this integration. The negative resistance effect amplifies the sensor’s response to hydrogen, enhancing sensitivity. When hydrogen gas interacts with the sensor material, it changes the local electrical environment, shifting the negative resistance region, which can be detected with higher precision than traditional resistance-based sensing mechanisms.

In contrast to the more commonly observed positive resistance changes, some hydrogen sensors exhibit a fascinating phenomenon known as negative resistance. This effect, characterized by a decrease in electrical resistance upon hydrogen exposure, arises from complex interactions between hydrogen and the sensing material at the atomic level [79]. The mechanism of negative resistance changes involves several interrelated processes. When hydrogen molecules encounter the sensor surface, they typically dissociate into atomic hydrogen. These atoms can then be absorbed into the material’s lattice, triggering a cascade of effects.

Firstly, absorbed hydrogen atoms often donate electrons to the sensing material, potentially increasing the density of charge carriers and thereby reducing electrical resistance (electron transfer). Secondly, the incorporation of hydrogen can modify the material’s electronic band structure, sometimes creating new conduction pathways or altering the Fermi level in ways that enhance conductivity (band structure modification). Lastly, hydrogen absorption usually causes lattice expansion, which, while increasing resistance in some materials, can actually enhance electron mobility in others, leading to decreased resistance (lattice expansion). Compared to their positive resistance counterparts, negative resistance sensors often offer advantages such as faster response times due to rapid electron transfer processes. The decrease in resistance can also be easier to measure in certain circuit configurations, potentially improving sensitivity. However, the choice between positive and negative resistance sensors ultimately depends on specific application requirements, including the range of hydrogen concentrations to be detected, operating conditions, and compatibility with existing systems.

This enhanced sensitivity is complemented by improved selectivity. By carefully tuning the negative resistance characteristics, the sensor becomes more selective to hydrogen over other gases, crucial for safety, environmental monitoring, and industrial processes where hydrogen detection is critical. Furthermore, negative resistance devices operate with lower power consumption, an advantageous feature for battery-operated flexible sensors, particularly in wearable applications that require long battery life.

The fabrication of flexible hydrogen sensors with negative resistance involves advanced techniques such as layer-by-layer assembly, nanopatterning, and material doping. Layer-by-layer assembly allows precise control of material deposition, creating the necessary conditions for negative resistance. Nanopatterning using nanolithography enhances the electrical properties at the nanoscale, while material doping introduces specific dopants into the sensor material to modulate its electrical properties, promoting negative resistance.

These flexible hydrogen sensors with negative resistance have a wide range of applications. In wearable technology, they enable continuous monitoring of hydrogen exposure in environments where hydrogen is used or produced. In industrial safety, they provide real-time detection of hydrogen leaks in chemical plants or hydrogen storage facilities. In environmental monitoring, they help track hydrogen levels in the atmosphere for pollution control. Additionally, in medical devices, they have potential applications in detecting hydrogen-related biomarkers in breath analysis. The phenomenon of negative resistance in flexible hydrogen sensors marks a significant advancement in sensor technology. By leveraging the unique properties of negative resistance, these sensors achieve higher sensitivity, selectivity, and efficiency. The integration of advanced materials and fabrication techniques further enhances their capabilities, making them suitable for a broad range of cutting-edge applications. As research and development continue, these sensors hold immense potential to revolutionize hydrogen detection and monitoring, offering enhanced performance and reliability across various fields.

The recent advancements in flexible hydrogen sensors with negative resistance performances have showcased various materials and configurations, each with unique attributes and performance metrics. Lee et al. demonstrated the use of cracked palladium films on an elastomeric substrate [60], achieving a notable hydrogen detection range with high sensitivity due to the increased surface area and the unique mechanical properties of the substrate. Specifically, the sensor achieved a sensitivity change of 4% under hydrogen exposure, demonstrating excellent flexibility and durability across 100–200 bending cycles. Similarly, Zhu et al. reported the development of Ti_3_C_2_T_x_ MXene@Pd colloidal nanocluster paper film (Figure 5) [59], which not only offered flexibility but also enhanced the sensitivity and response time of the hydrogen sensor, making it a promising candidate for practical applications. The sensor composed of MXene/Pd displayed a sensitivity range of 0.5–40% over 100–200 bending cycles, leveraging Pd colloidal dispersion for enhanced performance. In a different approach, Lim et al. utilized metal nanotubes synthesized via wet-chemical methods along sacrificial nanowire templates [62]. This method produced ultrasensitive, flexible chemical sensors, highlighting the potential for low-cost and scalable production. The Si/SiO_2_-Pd sensor was capable of detecting hydrogen concentrations between 100 to 10,000 ppm, maintaining performance over 1000–100,000 bending cycles due to its robust Pd nanotube structure.

Hao et al. explored the use of Pd-WS_2_/Si heterojunctions, which enabled highly sensitive hydrogen detection at room temperature (a sensitivity range of 0.1–4% H_2_, demonstrating stable operation over 1000–5000 bending cycles and 5–30 days of continuous use) (Figure 6A,B) [61], showcasing the advantage of utilizing heterojunctions for enhancing sensor performance. Cho et al. introduced a high-sensitivity, low-power flexible Schottky hydrogen sensor based on silicon nanomembrane (0.1–0.5% H_2_ and 1000–10,000 bending cycles) [80], which demonstrated excellent performance with low power consumption, making it ideal for integration into portable devices. Kuru et al. developed a high-performance flexible hydrogen sensor using a WS_2_ nanosheet-Pd nanoparticle composite film [81]. This composite film exhibited remarkable flexibility and high sensitivity, underscoring the synergy between WS_2_ and Pd nanoparticles in enhancing sensor performance (a detection range of 500–10,000 ppm H_2_ over 100 bending cycles on a PI substrate). Xie et al. fabricated Pd nanoparticle films on polymer substrates, with a sensitivity range of 130–760 ppm RH, capable of withstanding 500 bending cycles due to sputtering Pd on PET, resulting in transparent and flexible hydrogen sensors with significant potential for integration into electronic skin and other wearable devices [82]. Sun et al. further extended this work by electrodepositing Pd nanoparticles on single-walled carbon nanotubes, exhibiting a wide detection range of 100–10,000 ppm H_2_, withstanding 2000 bending cycles using the CVD technique to deposit SWNTs on PET. This contributes to the creation of flexible hydrogen sensors with high performance due to the unique properties of carbon nanotubes [51,52]. Ren et al. presented an ultrasensitive and wide-range flexible hydrogen sensor based on Pd-nanoparticle-decorated ultrathin SnO_2_ films (Figure 6C,D) [83]. This sensor exhibited a broad detection range and high sensitivity (with a detection range of 0.1–30 ppm RH, enduring 2000 bending cycles), which is crucial for various hydrogen sensing applications. Zhang et al. developed a flexible nanofiber sensor capable of detecting low concentrations of hydrogen, demonstrating the potential of nanofiber-based materials for sensitive hydrogen detection [84]. Shin et al. utilized flower-like palladium nanoclusters on graphene electrodes (Figure 7) [50], achieving ultrasensitive and flexible hydrogen gas sensing. This structure provided a large surface area and excellent electrical properties, contributing to the sensor’s high performance. Su et al. fabricated a flexible hydrogen sensor using Pd-nanoparticle-modified polypyrrole films, which showed promising results in terms of sensitivity and flexibility (a detection range of 100–5000 ppm H_2_) (Figure 8A,B) [85]. Shahzamani et al. explored palladium thin films on microfiber filtration paper as a flexible substrate, revealing a novel hydrogen gas sensing mechanism with good sensitivity and flexibility (a sensitivity range of 0.5–10% under bending conditions) (Figure 8C,D) [86]. Jang et al. developed hollow Pd-Ag composite nanowires for fast-responding and transparent hydrogen sensors, offering a unique combination of transparency and high performance (a detection range of 100–900 ppm H_2_) [87]. Finally, Zeng et al. utilized networks of ultrasmall palladium nanowires formed on filtration membranes (a sensitivity range of 0.01–0.3% H_2_) [88], achieving excellent hydrogen gas sensing performance due to the high surface-to-volume ratio and the efficient electron transport properties of the nanowires.

Overall, the advancements in flexible hydrogen sensors highlight the importance of material innovation and structural engineering in enhancing sensor performance. The comparative analysis of these studies reveals that while Pd-based materials dominate the field due to their excellent hydrogen absorption properties, the integration with other materials like WS_2_, carbon nanotubes, and MXene further improves the sensitivity, response time, and flexibility of the sensors. These developments pave the way for the future integration of flexible hydrogen sensors into a wide range of applications, from wearable devices to industrial monitoring systems. A comprehensive comparison of the performance characteristics of various flexible hydrogen sensors discussed in this section is presented in Table 1.

## 3. Stretchable Electronics for Hydrogen Sensors

The development and implementation of stretchable electronics for hydrogen sensing address critical needs driven by the increasing reliance on hydrogen as a clean energy source [48,89]. While flexible electronics have already made significant strides in this field, stretchable electronics represent a further evolution, providing additional benefits and capabilities [8,90]. Stretchable hydrogen sensors represent a significant leap forward in sensing technology. Unlike merely flexible sensors, stretchable sensors maintain functionality under extreme deformations, including stretching, twisting, and compressing [60,91]. This advancement opens up new possibilities for hydrogen detection in highly dynamic environments and on complex, non-planar surfaces [5].

Stretchable sensors offer several unique advantages over their flexible counterparts. They can conform to irregular and moving surfaces, making them ideal for wearable applications and integration into soft robotics [87,92]. Furthermore, their ability to withstand high strain while maintaining sensitivity to hydrogen allows for deployment in scenarios where traditional rigid or even flexible sensors would fail, such as in expandable storage tanks or on vehicle fuel lines that experience significant vibration and movement [80,93].

This section explores the unique fabrication techniques, such as intrinsically stretchable materials and engineered geometries, that enable these sensors to withstand high strain while maintaining high sensitivity to hydrogen [94,95]. We also discuss the challenges in achieving reliable electrical connections in stretchable systems and innovative solutions like liquid metal interconnects [73]. Additionally, we delve into the various strategies employed to integrate these highly deformable sensors into practical hydrogen detection systems, highlighting the merits and limitations of different sensor designs [88,96].

By overcoming the limitations of conventional sensors and leveraging the advanced properties of stretchable materials, these innovative devices offer enhanced sensitivity, integration versatility, and durability in extreme conditions [50,62]. This section elaborates on the device mechanisms and performance characteristics of stretchable electronics for hydrogen sensing, providing insights into their potential to revolutionize hydrogen safety monitoring in various applications.

### 3.1. Operation Mechanism of Stretchable Hydrogen Sensors

The sensing mechanism of stretchable hydrogen sensors, particularly those utilizing palladium (Pd) films, closely aligns with the principles employed in crack sensors. The core mechanism hinges on the formation and modulation of cracks or nanogaps within the Pd film when exposed to hydrogen gas [66].

Initially, palladium’s unique ability to absorb hydrogen plays a pivotal role (Figure 9A) [67]. Upon exposure to hydrogen, palladium forms palladium hydride (PdH_x_), leading to a significant increase in the lattice parameter of the Pd film. As the hydrogen concentration increases, a phase transition occurs in the Pd film, shifting from alpha-phase PdH_x_ (α-PdH_x_) to beta-phase PdH_x_ (β-PdH_x_). This transition significantly alters the electrical resistance of the Pd film. To maintain a linear relationship between hydrogen concentration and electrical resistance, sensors are designed to operate within a range that avoids this phase transition. The hydrogen-induced swelling causes the Pd film to expand. When this expansion occurs on an elastomeric substrate, it results in the formation of cracks or nanogaps due to the mechanical stress from the swelling (Figure 9B) [60,97,98]. The crack-based sensing mechanism involves the propagation and closure of cracks upon hydrogen exposure and desorption. Repeated exposure to hydrogen causes the cracks to widen due to continued Pd film expansion. Upon hydrogen desorption, the Pd film contracts, partially closing the cracks. This cyclic process modulates the electrical pathways in the Pd film, leading to measurable changes in resistance. The reversibility of crack formation and closure is critical for reliable sensor operation over many hydrogen exposure and removal cycles. The flexibility of the elastomeric substrate, such as polydimethylsiloxane (PDMS), is crucial as it can stretch and compress reversibly, maintaining the integrity of the sensor during repeated expansion and contraction cycles of the Pd film [99,100]. The use of elastomeric substrates enhances the sensor’s sensitivity by enabling larger and more controlled crack openings that directly correlate with hydrogen concentration.

Another mechanism of stretchable hydrogen sensors is based on strain engineering, a technique employed to create and control the formation of nanogaps (Figure 10) [101,102,103]. By applying mechanical strain or utilizing methods like liquid nitrogen freezing, the distribution and size of the cracks can be optimized, enhancing sensor performance. Techniques such as adjusting the initial hydrogen exposure concentration and heat treatment further refine the crack formation process [104], improving the sensor’s detection limits and sensitivity.

Overall, the combined effects of palladium’s hydrogen absorption properties, the mechanical-stress-induced cracking on elastomeric substrates, and advanced strain engineering techniques result in highly sensitive and reliable stretchable hydrogen sensors. Table 2 provides a detailed comparison of the performance characteristics of the stretchable hydrogen sensors discussed in this section. These sensors exhibit superior performance characteristics, including high sensitivity, rapid response, and excellent reversibility, making them ideal for a wide range of applications in hydrogen detection.

### 3.2. Stretchable Hydrogen Sensor Performance

Gurlo et al. have fabricated stretchable hydrogen sensors that utilize the unique properties of palladium films on compliant substrates to detect hydrogen with high sensitivity, rapid response times, and enhanced durability [65]. The core mechanism revolves around the reversible swelling of palladium when it absorbs hydrogen, leading to significant lattice expansion and the formation of nonstoichiometric hydrides, PdHx. This swelling is ingeniously harnessed by inducing controlled cracking in the Pd film. The sensors exploit the α to β phase transition of palladium, which occurs at higher hydrogen concentrations. This transition significantly increases the lattice parameter from approximately 3.90 Å in the α-phase to around 4.04 Å in the β-phase, enabling the detection of a broad range of hydrogen concentrations. These sensors exhibit high sensitivity, capable of detecting hydrogen concentrations as low as 0.01%. This sensitivity is due to the multiply cracked structure of the Pd film, which creates numerous nanogaps that respond even to minimal hydrogen-induced swelling. The electrical resistance of the Pd film changes linearly with hydrogen concentration within the α-phase region, providing a clear and quantifiable signal for hydrogen detection. The response time of these sensors is rapid, thanks to the fast kinetics of hydrogen absorption and desorption in the thin Pd films. The elastomeric substrate facilitates the swift mechanical adjustment of the Pd film during these processes, ensuring a prompt response to hydrogen presence. This design choice markedly improves the long-term stability and reliability of the sensors through the stretchable nature of the substrate preventing common issues such as spalling and delamination of the Pd film during the repeated expansion and contraction steps, making them a promising technology for reliable hydrogen detection.

Lee et al. demonstrated stretchable hydrogen sensors utilizing cracked palladium (Pd) films on elastomeric substrates (Figure 11) [60]. The fundamental mechanism is based on the reversible swelling of palladium when it absorbs hydrogen, which causes a substantial lattice expansion and the formation of nonstoichiometric hydrides (PdH_x_). This physical change is cleverly harnessed by inducing controlled cracking in the Pd film. The cracked Pd film on a PDMS substrate demonstrated reversible and significant On–Off responses under a wide range of hydrogen concentrations, with large current variations and a fast response time of less than 1 s. The performance of the cracked Pd film on the elastomeric substrate was compared to that on a Si/SiO_2_ substrate. The Pd film on the elastomeric substrate operated based on the crack open–close mechanism, whereas the Pd film on the Si/SiO_2_ substrate relied on electron scattering, showing slower and less sensitive responses. The use of elastomeric substrates, such as poly(dimethylsiloxane) (PDMS), is particularly important because it allows the Pd film to expand and contract without delaminating, a common issue with rigid substrates. The elastomeric substrate accommodates the mechanical strain from the Pd film, maintaining the sensor’s structural integrity through numerous cycles of hydrogen absorption and desorption. These sensors exhibit high sensitivity, capable of detecting hydrogen concentrations as low as 0.4%, which is attributed to the multiply cracked structure of the Pd film. The electrical resistance of the Pd film changes linearly with hydrogen concentration within the α-phase region, providing a clear and quantifiable signal for hydrogen detection. The sensors demonstrate a rapid response time, achieving significant resistance changes within less than 1 s upon exposure to hydrogen because of the fast kinetics of hydrogen absorption and desorption in the thin Pd films.

Zhao et al. prepared a stretchable hydrogen sensor with palladium (Pd) nanogap arrays fabricated through a sophisticated swelling-induced cracking method (Figure 12) [91]. The fabrication process begins with the preparation of an epoxy layer, which is spin-coated onto a glass substrate and solidified to achieve a thickness of 40 μm. Following this, a positive photoresist is applied and patterned using photolithography. This is followed by the deposition of chromium (Cr) and palladium (Pd) films, with thicknesses of 30 nm and 90 nm, respectively. The next critical step involves immersing the patterned epoxy in ethanol, which causes the epoxy to swell. This swelling induces controlled cracking in the Pd film at the pre-defined stress concentrative points, forming uniform nanogaps. This process results in a periodic bulging effect at the necking areas of the electrodes, with a height difference of about 400 nm, which is sufficient to crack the Pd film and form nanogaps approximately 200 nm wide. The performance of the hydrogen sensors, which were fabricated on both epoxy and polyimide (PI) substrates, is impressive. For the epoxy substrate, the sensor demonstrates a detection limit of 400 ppm when the average nanogap width is 68 nm. This detection limit increases to 7000 ppm when the nanogap width is increased to 112 nm. The response time for this configuration is 32 s, with a recovery time of 21 s at a hydrogen concentration of 400 ppm. In contrast, the hydrogen sensors fabricated on PI substrates exhibit even lower detection limits, thanks to the smaller average nanogap width of approximately 10 nm. These sensors show a detection limit of less than 200 ppm. The response time for the PI-based sensors is 116 s, and the recovery time is 143 s for hydrogen concentrations ranging from 200 to 10,000 ppm. The performance of the PI-based sensors remains stable up to a temperature of 180 °C. However, post-annealing tests at this temperature reveal a decrease in current variation by 40–44%, and a slight increase in response and recovery times due to the thermal expansion mismatch between the PI substrate and Cr/Pd electrodes. Moreover, over multiple cycles, the sensors show only a 3.8% variation in resistance change after 15 cycles, underscoring the robustness and reliability of the nanogap arrays.

The mechanism of action of stretchable hydrogen sensors relies on the interplay between hydrogen-induced palladium film expansion, crack formation, and the reversible mechanical properties of elastomeric substrates. This mechanism allows for highly sensitive and reliable detection of hydrogen gas through measurable changes in electrical resistance. Compared to conventional Pd-based hydrogen sensors, which often suffer from mechanical degradation and limited sensitivity at low hydrogen concentrations, these stretchable sensors offer substantial improvements. The combination of the compliant substrate and the innovative use of crack formation and closure mechanisms result in superior performance metrics, making them a promising technology for reliable hydrogen detection.

In real-world practical applications, the performance of flexible hydrogen sensors is critical. Studies have shown that these sensors can maintain their performance under varying temperature and humidity conditions [108]. For example, graphene-based flexible hydrogen sensors have demonstrated stable performance in the temperature range of −50 °C to 100 °C [109], which is suitable for practical applications. Long-term stability is another crucial factor. Extended durability tests have shown that some flexible sensors can maintain their sensitivity for over 1000 h of continuous operation [110]. This is particularly important for sensors integrated into vehicles or stationary infrastructure for practical applications such as monitoring and safety systems, and fuel cell technology. In terms of response time, flexible sensors based on Pd nanoparticles have achieved response times as low as 1 s for a 1% hydrogen concentration, which is faster than many conventional rigid sensors [53,111].

More specifically, in portable hydrogen detection devices, used by maintenance workers in hydrogen facilities, chemiresistive sensors based on Pd nanoparticles have shown promise [8,112,113]. These sensors offer a combination of flexibility, fast response times, and high sensitivity, making them ideal for personal safety equipment. Another emerging application is in smart textiles for hydrogen facility workers. Flexible sensors based on 2D materials have been integrated into fabrics [114,115], creating wearable safety devices that can alert workers to hydrogen leaks while being comfortable and unobtrusive. For comprehensive monitoring of large hydrogen infrastructure, distributed sensor networks using various flexible sensor types have been proposed [116]. These networks can provide real-time, wide-area hydrogen detection, crucial for the safe operation of hydrogen mobility infrastructure.

### 3.3. Challenges in Stretchable Hydrogen Sensors

While flexible and stretchable hydrogen sensor technologies hold great promise, several significant challenges must be overcome for practical applications. This section will examine these key challenges and discuss the approaches researchers are currently taking to address them. The first major challenge is sensor reliability under high-strain conditions. Flexible and stretchable sensors, by their nature, can be subjected to various forms of deformation. Won et al. [105] developed ultrasensitive stretchable conductive fibers using percolated Pd nanoparticle networks, which maintained hydrogen sensing capabilities even when stretched up to 100% strain. Additionally, Kim et al. [63] developed a high-resolution, fast, and shape-conformable hydrogen sensor platform using polymer nanofiber yarn coupled with nanograined Pd@Pt, which demonstrated stable performance under various bending conditions. The second crucial challenge is differentiating between strain-induced resistance changes and hydrogen-induced resistance changes. This is vital for the accuracy and reliability of stretchable hydrogen sensors. Namgung et al. [106] created sensors that could distinguish between these two effects by transferring palladium nanosheets onto an elastomeric substrate. They incorporated a reference electrode to compensate for strain effects. Similarly, Lee et al. [60] developed cracked palladium films on an elastomeric substrate, utilizing unique crack formation and propagation characteristics to differentiate between mechanical deformation and hydrogen exposure effects. Long-term stability and fatigue resistance represent the third major challenge. Lim et al. [62] investigated the durability of their flexible chemical sensors over thousands of bending cycles, highlighting the importance of maintaining sensor performance under repeated deformation. The fourth challenge relates to temperature effects on flexible and stretchable sensors. Temperature variations can significantly impact sensor performance, particularly in relation to their flexible and stretchable nature.

## 4. Conclusions

This comprehensive review has elucidated the significant advancements in flexible and stretchable electronics for hydrogen sensing applications. The field has witnessed remarkable progress, particularly in the development of highly sensitive and responsive sensors based on innovative material combinations such as cracked palladium films on elastomeric substrates and palladium nanoparticle films on polymer substrates. These cutting-edge sensors have demonstrated exceptional capabilities, with some designs achieving hydrogen detection limits as low as 0.01% and exhibiting rapid response times. The implementation of advanced fabrication techniques, including layer-by-layer assembly and nanopatterning, has played a pivotal role in enhancing the overall performance and reliability of these sensors. Looking toward the future, the trajectory of flexible and stretchable hydrogen sensors presents immense potential for revolutionizing hydrogen safety protocols and expanding the utility of hydrogen-based technologies. The ongoing evolution of this field is expected to address several critical challenges, with research efforts likely to focus on enhancing the long-term stability of sensors under diverse environmental conditions, improving selectivity in complex gas mixtures, and developing self-powered sensor systems for autonomous operation. Moreover, the integration of these advanced sensors with wireless communication technologies and Internet of Things (IoT) platforms represents a promising avenue for establishing real-time, distributed hydrogen monitoring networks.

The realization of these ambitious goals necessitates a multidisciplinary approach, calling for close collaboration among materials scientists, electrical engineers, and chemical engineers. Key areas that warrant further investigation include the optimization of sensor stability across a wide range of environmental parameters, the development of more sophisticated gas discrimination mechanisms to enhance hydrogen selectivity, the design of energy harvesting or low-power technologies for self-sustained sensor operation, the seamless integration of sensors with wireless communication protocols, and the scaling up of manufacturing processes to enable cost-effective, large-scale production.

As research in this domain progresses, significant advancements in sensor miniaturization, energy efficiency, and multifunctionality are anticipated. The convergence of artificial intelligence and machine learning algorithms with these advanced sensor technologies holds the promise of creating more intelligent and adaptive hydrogen detection systems, capable of real-time data analysis and predictive maintenance. By addressing the challenges related to scalability, system integration, and real-world performance, flexible and stretchable hydrogen sensors are poised to make substantial contributions to the advancement of hydrogen energy systems and safety applications. The potential applications of this technology span a wide range of sectors, including automotive, aerospace, industrial processes, and consumer applications, underscoring its transformative potential in enabling the safe and efficient utilization of hydrogen as a clean energy carrier.

The rapidly evolving field of flexible and stretchable electronics for hydrogen sensing offers unprecedented opportunities for enhancing safety and efficiency in the burgeoning hydrogen economy. As research continues to push the boundaries of materials science and sensor technology, these innovative sensors are expected to play a crucial role in facilitating the widespread adoption of hydrogen as a sustainable energy source. The future of this field holds great promise, with ongoing advancements poised to overcome current limitations and unlock new possibilities in hydrogen sensing and beyond.

## 5. Future Perspectives

The future of flexible and stretchable hydrogen sensors lies in the development and integration of advanced materials. Nanomaterials such as graphene, carbon nanotubes, and metal–organic frameworks offer exceptional properties that can significantly enhance sensor performance. Future research should prioritize the development of multi-functional nanocomposites. By combining the unique properties of various nanomaterials, sensors can achieve enhanced sensitivity, selectivity, and durability. Innovations in scalable synthesis methods, such as chemical vapor deposition (CVD) and roll-to-roll processing, will enable the large-scale production of high-performance nanomaterials, making advanced sensors more accessible and affordable.

Improving key performance metrics is essential for the future viability of hydrogen sensors. Sensitivity and selectivity can be significantly enhanced by utilizing catalytic materials and surface modifications, which will help achieve high selectivity for hydrogen over other gases and reduce cross-sensitivity issues. Additionally, optimizing sensor design and material properties to facilitate faster adsorption and desorption of hydrogen molecules will result in sensors with rapid response and recovery times, crucial for real-time monitoring. Ensuring that sensors maintain high performance under mechanical deformation is also vital. Research should focus on developing materials and structures capable of withstanding repeated bending, stretching, and twisting without degradation.

The integration of hydrogen sensors with the Internet of Things (IoT) and smart systems represents a significant future direction. Incorporating wireless communication capabilities into sensors will enable real-time data transmission and remote monitoring, enhancing usability in industrial and environmental applications. Developing self-powered sensors that can harvest energy from the environment, such as through piezoelectric or photovoltaic methods, will ensure long-term operation without the need for frequent battery replacements. Additionally, creating networks of sensors that can communicate and process data collectively will improve the accuracy and reliability of hydrogen detection, especially in large-scale industrial settings.

The future will see hydrogen sensors being deployed in a variety of novel applications. Flexible hydrogen sensors integrated into wearable devices will provide continuous monitoring for personnel working in hydrogen-rich environments, significantly enhancing safety. In hydrogen-powered vehicles, sensors embedded in fuel cells and storage tanks will ensure the safe operation of these vehicles by detecting leaks and monitoring hydrogen purity. Deploying networks of hydrogen sensors in strategic locations for environmental monitoring will help track atmospheric hydrogen levels, contributing to pollution control and environmental protection.

To facilitate the widespread adoption of hydrogen sensors, establishing standards and regulations is imperative. Developing standardized testing protocols and performance benchmarks will ensure that sensors meet the required sensitivity, selectivity, and durability criteria. Implementing safety guidelines for the deployment of hydrogen sensors in various environments will ensure that they effectively contribute to hazard prevention and management.

Looking ahead, several key areas of development are poised to significantly advance flexible and stretchable hydrogen sensors for next-generation applications. Advancements in nanotechnology and microfabrication techniques are driving the miniaturization of sensors, potentially to the nanoscale, enabling more comprehensive and granular hydrogen monitoring in vehicles and infrastructure. Concurrently, efforts to improve energy efficiency may lead to the development of self-powered sensors leveraging energy harvesting technologies, crucial for wearable applications and remote deployments. The integration of these sensors with the IoT will allow for real-time, wireless monitoring across entire mobility ecosystems, while the development of multi-functional sensors could combine hydrogen detection with other critical measurements such as temperature or pressure. Enhancing the durability and reliability of these sensors under real-world conditions remains a key challenge, with research focusing on new protective coatings and encapsulation techniques. To support widespread adoption, advancements in large-scale manufacturing techniques, such as roll-to-roll processing and 3D printing, will be vital. Innovative approaches, including bio-inspired designs, might lead to self-healing sensors capable of recovering from mechanical damage, significantly extending their lifespan in harsh environments. Finally, as the field matures, the development of international standards and regulatory frameworks will be crucial to ensure reliability and interoperability across different hydrogen mobility platforms and infrastructure.

While our review has focused on the recent advancements in flexible and stretchable hydrogen sensors, an intriguing area for future development lies in the detection and differentiation of hydrogen isotopes. Recent studies have shown promising results in distinguishing between hydrogen isotopes, particularly protium and deuterium, using various sensing technologies [117,118,119,120,121]. However, these isotope-sensitive detection methods have not yet been integrated into flexible and stretchable sensor designs. This gap presents an exciting opportunity for future research and development. The integration of isotope-sensitive detection mechanisms into flexible and stretchable hydrogen sensors could significantly expand their capabilities and application range, particularly in fields such as nuclear energy, fusion research, and isotope tracing. The development of such advanced sensors would require interdisciplinary collaboration, bringing together expertise in materials science, sensor technology, and isotope chemistry. While challenging, this direction of research has the potential to open new avenues in fields such as nuclear safety, fusion energy research, and environmental monitoring.

The future of flexible and stretchable hydrogen sensing is promising, with significant advancements expected in material science, sensor integration, and application domains. By addressing the challenges of performance, scalability, and integration with smart systems, researchers and industry professionals can unlock the full potential of these innovative sensors, paving the way for a safer and more sustainable hydrogen economy. Interdisciplinary collaboration and continued investment in research and development will be key to realizing these advancements.

## Figures and Tables

**Figure 1 sensors-24-06637-f001:**
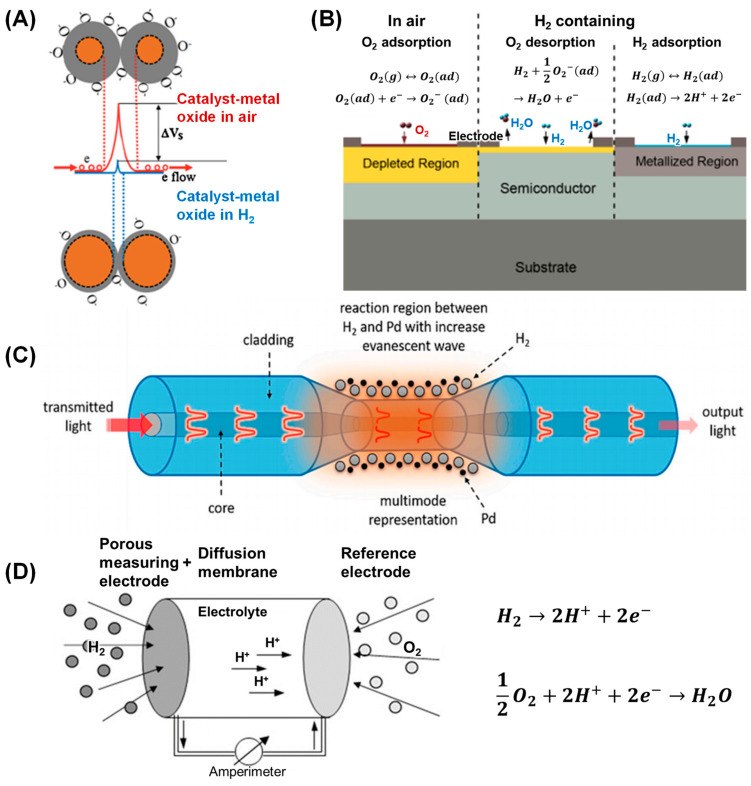
Schemes of different working mechanisms of hydrogen sensors: (**A**) Catalytic (reprinted with permission from Ref. [7]. Copyright 2013 Elsevier). (**B**) Resistive (reprinted from Ref. [14]; copyright 2012 MDPI). (**C**) Optical (reprinted from Ref. [11]; copyright 2020 MDPI). And (**D**) electrochemical (reprinted with permission from Ref. [13]; copyright 2013 Elsevier).

**Figure 2 sensors-24-06637-f002:**
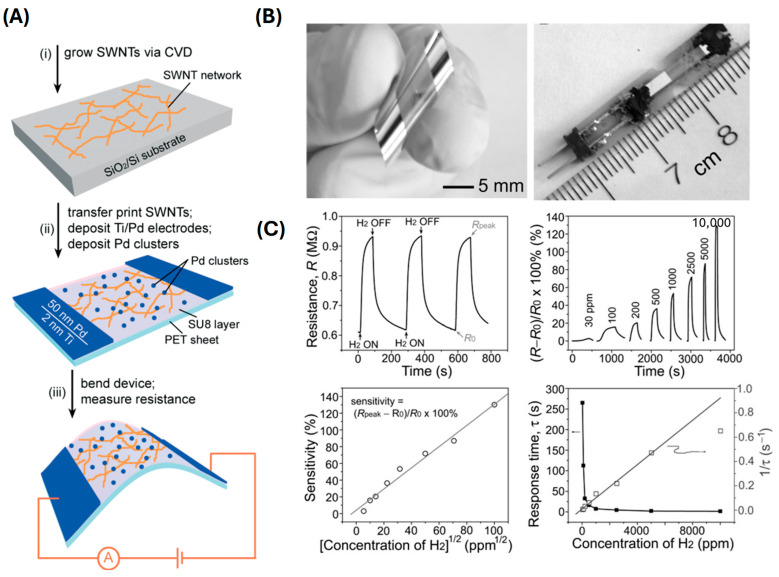
(**A**) Schematic of the process for fabricating flexible hydrogen sensor by using a film of SWNTs decorated with Pd nanoparticles on a plastic SU8/PET substrate nanoparticle. (**B**) Optical and AFM images of the fabricated flexible sensors. (**C**) Evaluation of the performance of a sensor for sensing hydrogen molecules in air at room temperature. (Reprinted with permission from Ref. [73]. Copyright 2007 Wiley).

**Figure 3 sensors-24-06637-f003:**
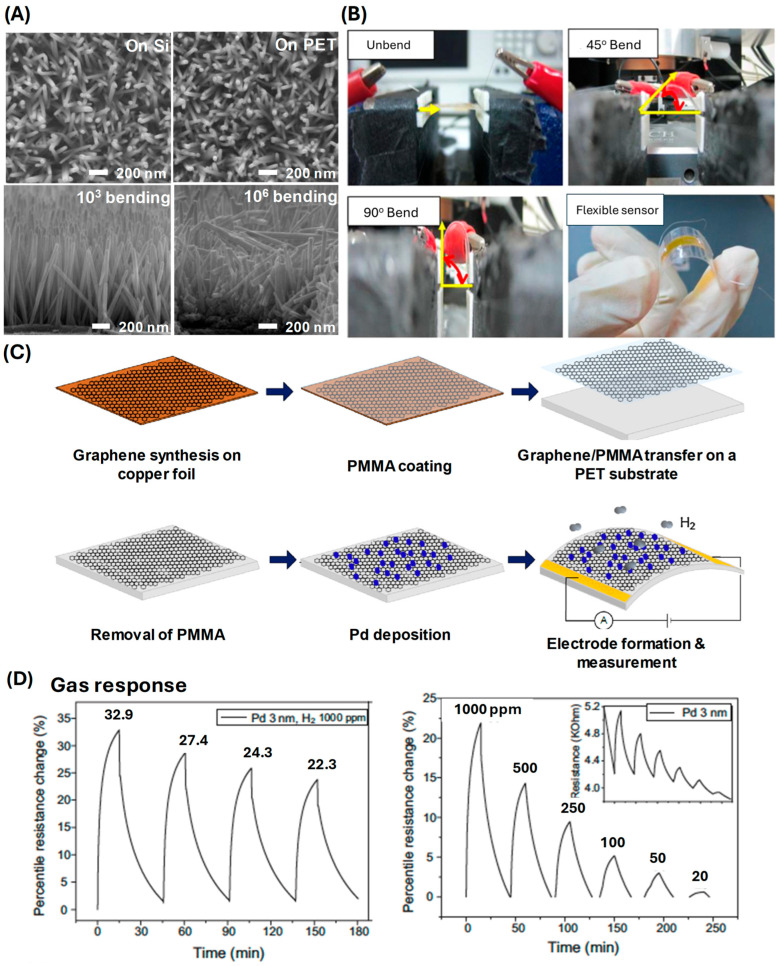
(**A**) SEM images of Pd-loaded ZnO NRs on PI and flexible polyester (PET). (**B**) Optical images depending on the bending angle 0~90° (reprinted with permission from Ref. [53]; copyright 2013 Elsevier). (**C**) Schematic of the fabrication procedure of the H_2_ gas sensor. (**D**) Sensing performance of the graphene sensor (reprinted with permission from Ref. [74]; copyright 2012 Elsevier).

**Figure 4 sensors-24-06637-f004:**
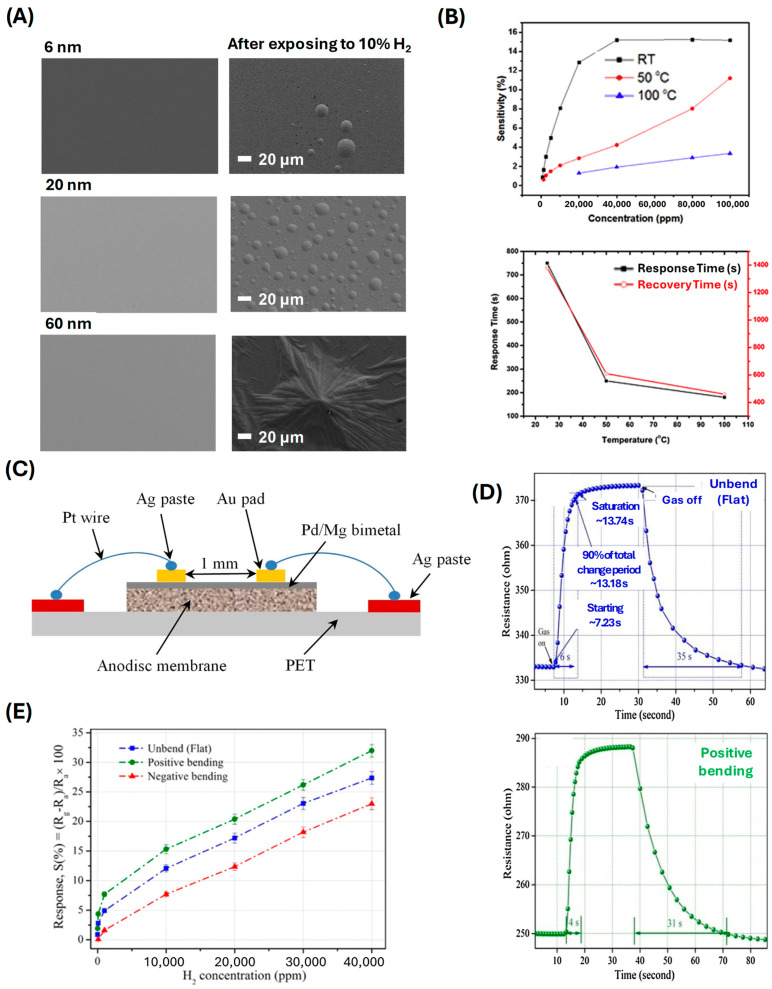
(**A**) SEM images of Pd thin films coated to various thicknesses on glass substrate. (**B**) H_2_ sensing sensitivity and response time depending on temperature (reprinted with permission from Ref. [75]; copyright 2016 Elsevier). (**C**) Schematic of Pd/Mg bimetallic nanowires H_2_ sensor. (**D**) Response–recovery time characteristics depending on bending condition. (**E**) Response variation for various H_2_ concentrations depending on bending conditions (reprinted with permission from Ref. [76]; copyright 2017 Elsevier).

**Figure 5 sensors-24-06637-f005:**
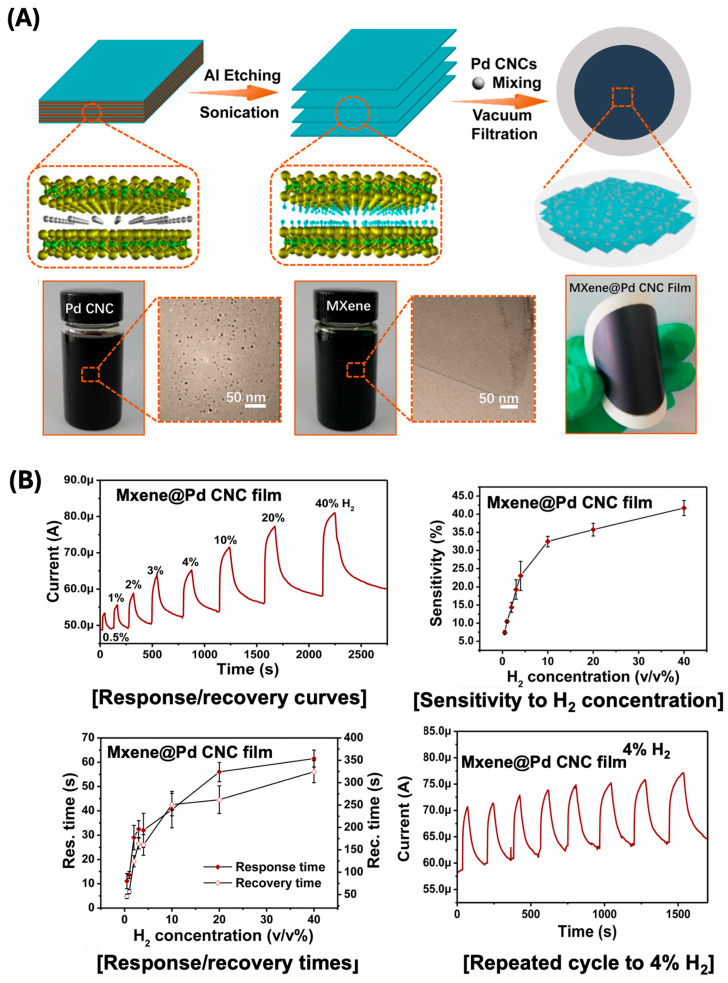
(**A**) Fabrication schematic of MXene and MXene@Pd CNC films. (**B**) H_2_ sensing sensitivity and response/recovery time as a function of H_2_ concentration (reprinted with permission from Ref. [59]; copyright 2020 Elsevier).

**Figure 6 sensors-24-06637-f006:**
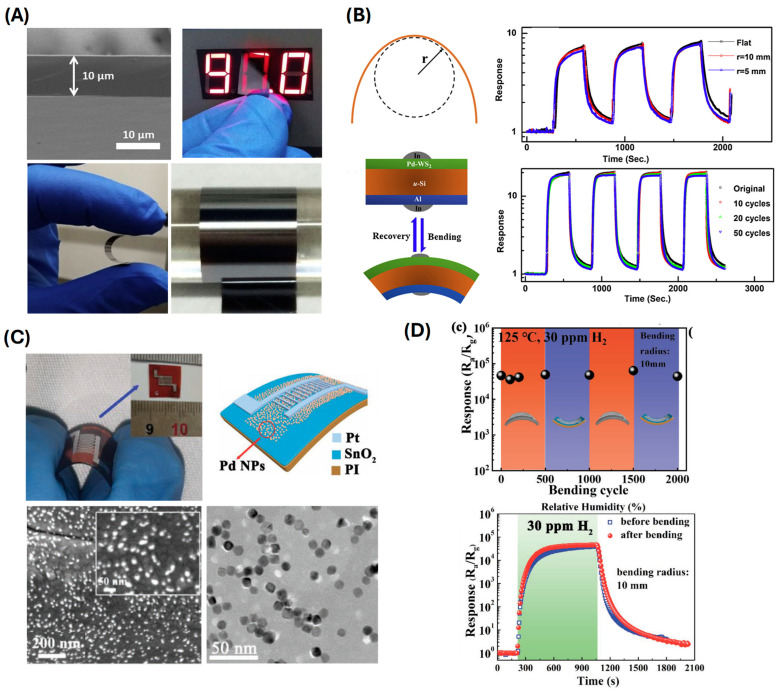
(**A**) Optical image of u-Si substrate. (**B**) Dynamic response depending on bending condition and bending cycle (reprinted with permission from Ref. [61]; copyright 2019 Elsevier). (**C**) Photo and schematic image for the structure of Pd NPs/SnO_2_/PI sensor. (**D**) Sensing stability of Pd NPs/SnO_2_/PI sensor after bending. (reprinted from Ref. [83]; copyright 2023 Wiley).

**Figure 7 sensors-24-06637-f007:**
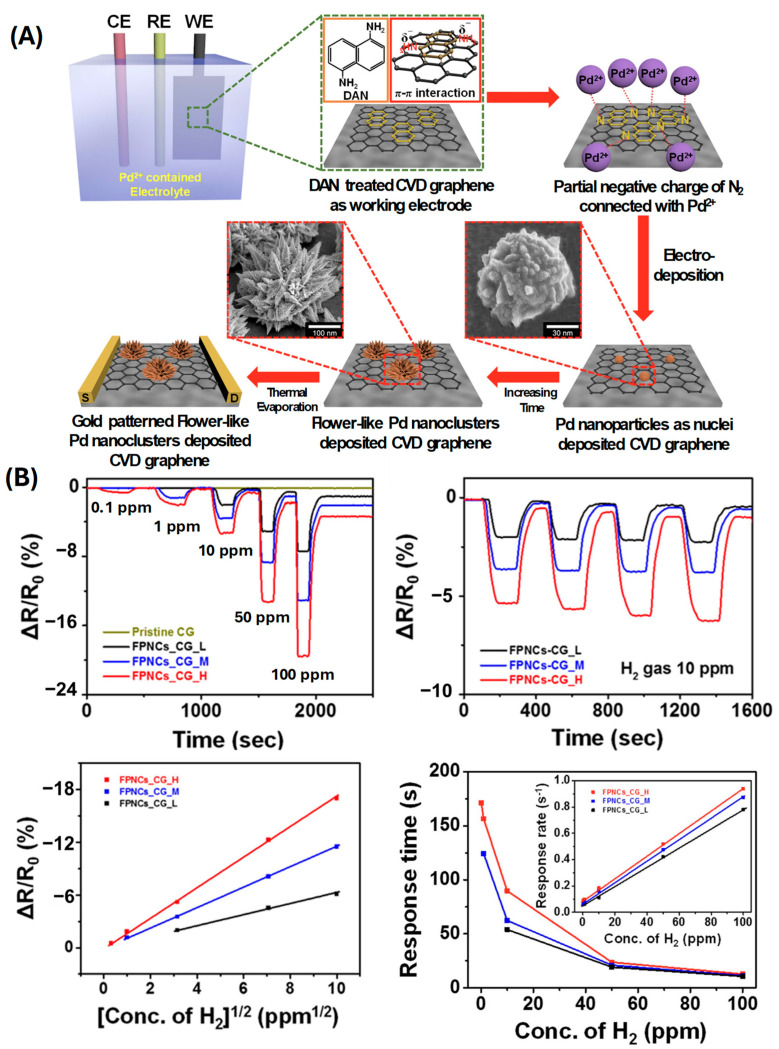
(**A**) Schematic for the fabrication of FPNCs_CG electrode. (**B**) Sensing selectivity and response time depending on H_2_ concentration (reprinted from Ref. [50]; copyright 2015 Springer Nature).

**Figure 8 sensors-24-06637-f008:**
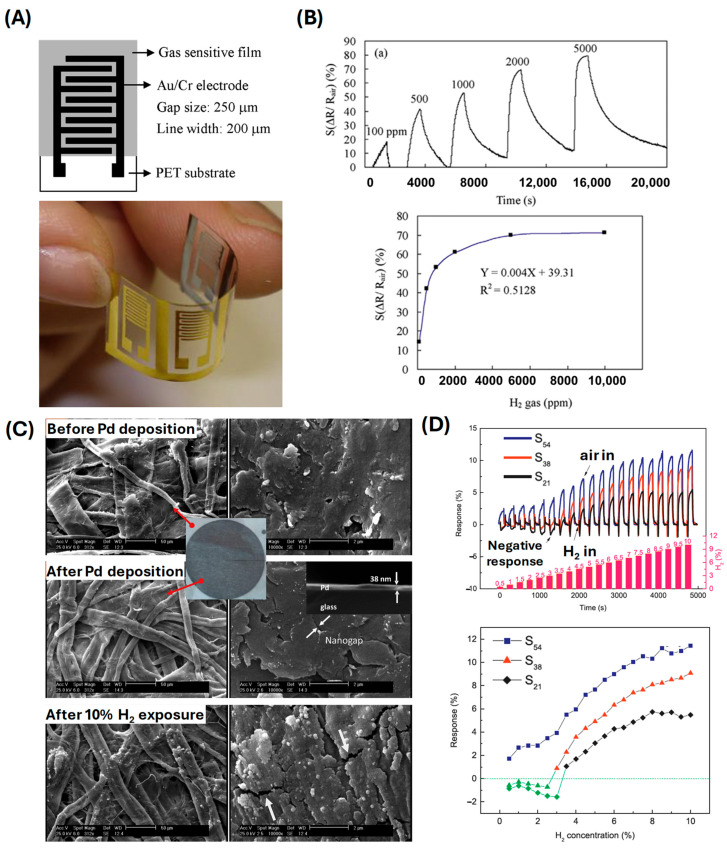
(**A**) Schematic and optical image of Pd NPs-PPy on PET H_2_ sensor. (**B**) Response of H_2_ sensor for various H_2_ concentration cycles (reprinted with permission from Ref. [85]; copyright 2016 Elsevier). (**C**) SEM image of Pd on filtration paper H_2_ sensor. (**D**) Response curve and sensitivity of the filtration paper H_2_ sensor at different H_2_ concentrations for deposition thickness (reprinted with permission from Ref. [86]; copyright 2019 Elsevier).

**Figure 9 sensors-24-06637-f009:**
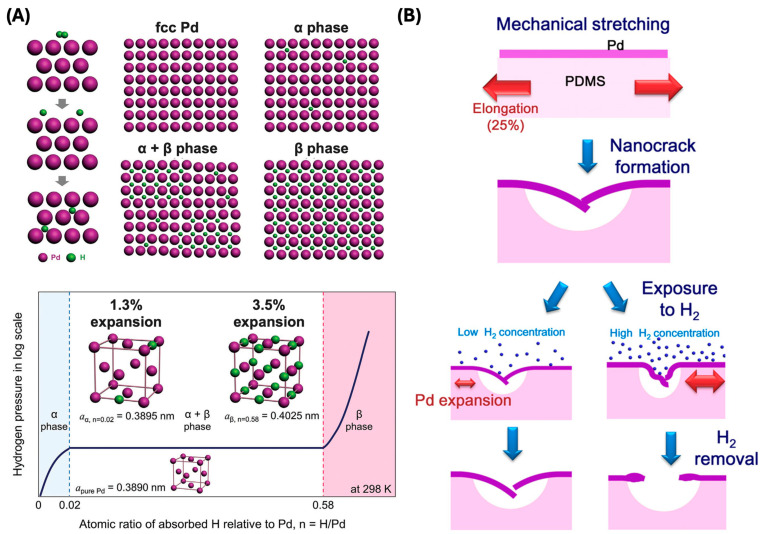
(**A**) Reaction steps between H_2_ molecule and Pd during H_2_ absorption (reprinted with permission from Ref. [67]; copyright 2013 Elsevier). (**B**) Forming nanogap on Pd/PDMS film using H_2_ exposure (reprinted with permission from Ref. [97]; copyright 2021 Wiley).

**Figure 10 sensors-24-06637-f010:**
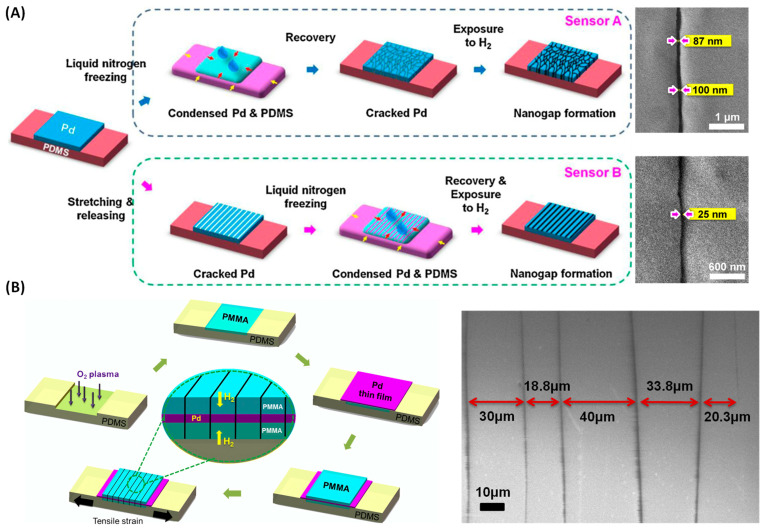
(**A**) Schematic fabrication processes and SEM images of nanogaps for two types of sensors: Sensor A, where no pre-strain to the Pd/PDMS is shown, and Sensor B, where the Pd/PDMS is elongated by 25% before LNF (reprinted with permission from Ref. [101]; copyright 2013 Elsevier). (**B**) Schematic of the fabrication of nanogap sensors using PMMA/Pd/PMMA on a PDMS substrate and SEM image of the cracks formed in PMMA-Pd-PMMA/PDMS (reprinted with permission from Ref. [102]; copyright 2014 Elsevier).

**Figure 11 sensors-24-06637-f011:**
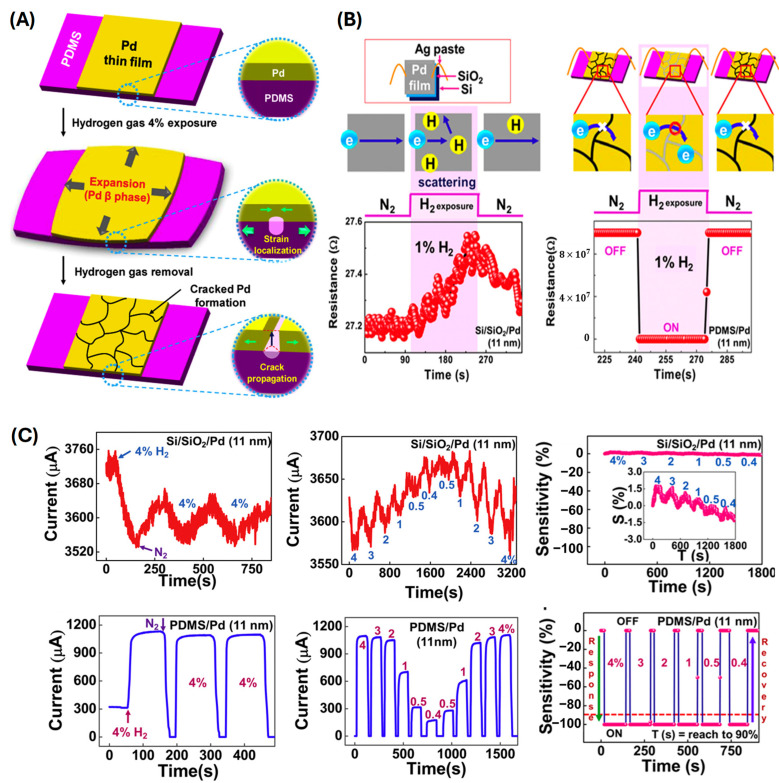
(**A**) Schematic of the fabrication cracked Pd on PDMS by H_2_ exposure. (**B**) Comparison of response time and respective mechanisms of a Pd-Si/SiO_2_ and Pd-PDMS sensor. (**C**) Response and sensitivity of a Pd thin film on a Si/SiO_2_ substrate and a PDMS for varying concentrations of H_2_ (reprinted with permission from Ref. [60]; copyright 2012 Elsevier).

**Figure 12 sensors-24-06637-f012:**
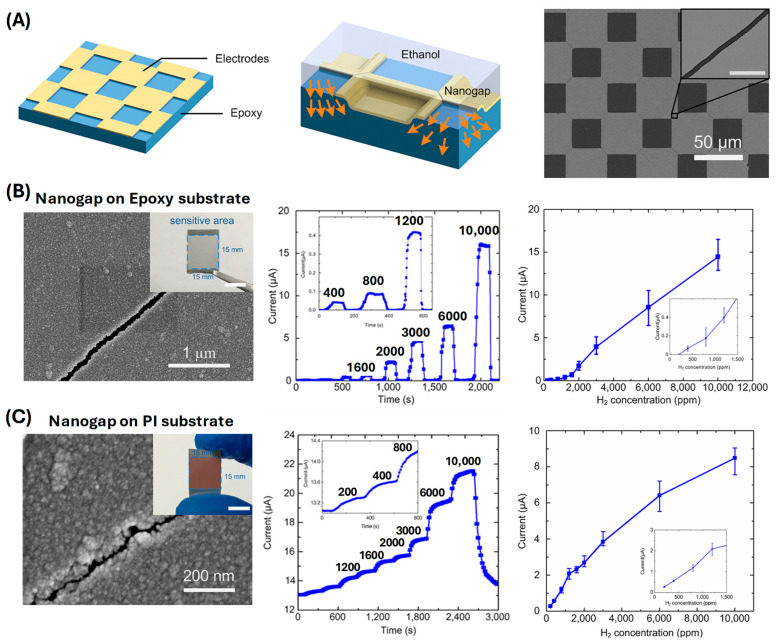
(**A**) Schematic illustration of the fabrication steps of the nanogap electrode. The arrows indicate the directions of epoxy expansion, which causes the cracking of the electrode and leads to the formation of nanogaps. (**B**) Detection limit of nanogap electrodes on epoxy substrate for gap width and edge length. (**C**) Performance of nanogap electrodes on PI substrate (reprinted with permission from Ref. [91]; copyright 2018 Elsevier).

**Table 1 sensors-24-06637-t001:** Comparison of performance characteristics of flexible hydrogen sensors.

Mechanism	Material	Hydrogen Sensing	Durability Test	Fabrication	Ref.
Negative resistance	polyethylene naphthalate film	0.1–100 ppm	-	Pd on graphene	[50]
	PET SWNT	100–10,000 ppm	2000 cycles	SWNT on PET via CVD technique	[51]
	PET SWNT	0.01–1%	Bending 1000 cycles	SWNT on PET via CVD technique	[52]
	Mxene/Pd	0.5–40%	Bending 100–2000 cycles	Film using Pd colloidal dispersion	[59]
	PDMS/Pd	4%	-	Crack (Hydrogen exposure	[60]
	Si/WS_2_-Pd	0.1–4% H2	Bending 1000–5000 cycles 5–30 days	DC sputtering WS_2_ on Ultrathin Si	[61]
	Si/SiO_2_	100–10,000 ppm depending on bending radius	Bending 1000–100,000 cycles	Pd Nanotube (wet chemical synthesis)	[62]
	Silicon nanomembrane	0.1–0.5%	Bending 1000–10,000 cycles	-	[80]
	WS_2_ Nanosheet Pd Nanoparticle	500–10,000 ppm	100 cycles	WS2-Pd solution bake on PI	[81]
	PET	130–760 ppm RH	500 cycles	Sputtering Pd on PET	[82]
	PI SnO2 Pd	0.1–30 ppm RH	Bending 2000 cycles	-	[83]
	polyacrylonitrile nanofiber network	2–50 ppm 200–800 ppm TEMP	-	PAN soaked in K2PdCl4 solution and reducted	[84]
	polypyrrole films	100–5000 ppm bending	-	-	[85]
	microfiber filtration paper	0.5–10% bending	-	Pd sputtering on filtration paper Crack (exposure hydrogen)	[86]
	Hollow Pd-Ag composite nanowire on polyimide	100–900 ppm	-	Deposition Ag on Ni/PI and galvanic replacement Pd depending on time Nano gap 10 μm	[87]
	Palladium nanowires formed on filtration membrane	0.01–3%	-	-	[88]
Positive resistance	PET	100–1000 ppm	Bending 100–100,000 cycles	Sputtering Pd NP on ZnO Transfer on PET	[53]
	PET	30–10,000 ppm	Bending 1000 cycles	SWNT/Ti/Pd electrode/Pd cluster	[73]
	PET	20–1000 ppm	-	Pd deposition on graphene	[74]
	PI	1250 ppm—10%	-	Sputtering Pd on PI	[75]
	Mesh of ultrasmall Pd/Mg bimetallic nanowire on PET	10,000 ppm Depending on bending direction (positive and negative)	-	Pd/Mg bimetal on PET	[76]

**Table 2 sensors-24-06637-t002:** Comparison of performance characteristics of stretchable hydrogen sensors.

Mechanism	Material	Hydrogen Sensing	Durability Test	Fabrication	Ref.
Negative resistance	Nanofiber yarn	100 ppm—4%	700 cycles	Pd was sputtered on the nano fiber, forming a nanogap between nanofibers	[60]
	PDMS	0.4–4%	-	Crack (H_2_ exposure)	[63]
	Epoxy	200–10,000 ppm	-	Patterned epoxy and then the electrode cracked after being immersed in ethanol	[91]
	PMMA/Pd/PMMA on PDMS	0.1–10%	-	PMMA/Pd/PMMA trilayer Crack (25% mechanical stretching 30 um gap size)	[102]
	Fiber yarn	5 ppm—10%	-	Crack (strain the fiber)	[105]
	PDMS/Pd	20–100 ppm	-	Crack (PDMS stretch)	[106]
	PDMS/Pd	0.05–10%	-	Crack (strain and hydrogen exposure)	[107]
Pd Crack sensor	Cyanoacrylate film	0.5–10% H_2_	-	Pd meso wire	[66]

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
