# Peer review of "Advancements in Flexible and Stretchable Electronics for Resistive Hydrogen Sensing: A Comprehensive Review"

_sensors, 2024, doi:10.3390/s24206637_

Round 1

Reviewer 1 Report

Comments and Suggestions for Authors

Advantages:

The author effectively summarized the recent progress in flexible/stretchable hydrogen sensing devices, providing a comprehensive overview

Limitations

1) The author briefly mentioned the hydrogen sensing principles of optical, electrochemical, and resistive type sensors. However, the author largely focused on the resistive/flexible sensors in this review. I would recommend that the author should clarify in the review/abstract and in the title, that the the main scope of this review is on the resistive type sensor. 

2) The abstract mentions that the review covers the challenges of stretchable and flexible sensors, but these challenges are not highlighted in detail in the main paper. It would strengthen the review to discuss certain challenges such as: Are sensors reliably able to detect under highly strained and deformed? Can sensors differentiate resistance between the strain-induced resistance change and the change in resistance due to hydrogen absorption?

3. It seems most of hydrogen sensors that the author described use palladium. I would suggest describing the sensing mechanism of hydrogen first before mentioning the positive and negative response. The current version describes the mechanism in the one of the last part. 

4. Although the mechanism with positive-response was well described, the negative sensing mechanism is not well explained 

5. The author mostly focused on Pd-based sensors. It would be great if the author rationalize why Pd-based sensors are mostly mentioned in this type of applications. A short and brief discussion on the material’s advantages and why such material based sensor is favored over others in such applications would improve the review

After addressing this comments, the review will be more focused and well-structured.

Author Response

Advantages:

The author effectively summarized the recent progress in flexible/stretchable hydrogen sensing devices, providing a comprehensive overview

After addressing these comments, the review will be more focused and well-structured.

Response: We would like to thank referees for their encouraging evaluation and constructive comments on our manuscript. In accordance to comments/suggestions from reviewers, careful revision has been made. The revised parts were highlighted in red.

Limitations

Comment 1. The author briefly mentioned the hydrogen sensing principles of optical, electrochemical, and resistive type sensors. However, the author largely focused on the resistive/flexible sensors in this review. I would recommend that the author should clarify in the review/abstract and in the title, that the main scope of this review is on the resistive type sensor.

Response: We would like to thank referees for their encouraging evaluation and constructive comments on our manuscript. According to the reviewer’s suggestion, we revised the title and abstract as follows:

Title

Advancements in Flexible and Stretchable Electronics for Resistive Hydrogen Sensing: A Comprehensive Review

Abstract

Flexible and stretchable electronics have emerged as a groundbreaking technology with wide-ranging applications, including wearable devices, medical implants, and environmental monitoring systems. Among their numerous applications, hydrogen sensing represents a critical area of research, particularly due to hydrogen's role as a clean energy carrier and its explosive nature at high concentrations. This review paper provides a comprehensive overview of the recent advancements in flexible and stretchable electronics tailored for resistive hydrogen sensing ap-plications. It begins by introducing the fundamental principles underlying the operation of flexible and stretchable resistive sensors, highlighting the innovative materials and fabrication techniques that enable their exceptional mechanical resilience and adaptability. Following this, the paper delves into the specific strategies employed in the integration of these resistive sensors into hydrogen detection systems, discussing the merits and limitations of various sensor designs, from nanoscale transducers to fully integrated wearable devices. Special attention is paid to the sensitivity, selectivity, and operational stability of these resistive sensors, as well as their performance under real-world conditions. Furthermore, the review explores the challenges and opportunities in this rapidly evolving field, including the scalability of manufacturing processes, the integration of resistive sensor networks, and the development of standards for safety and performance. Finally, the review concludes with a forward-looking perspective on the potential impacts of flexible and stretchable resistive electronics in hydrogen energy systems and safety applications, underscoring the need for interdisciplinary collaboration to realize the full potential of this innovative technology.

Comment 2. The abstract mentions that the review covers the challenges of stretchable and flexible sensors, but these challenges are not highlighted in detail in the main paper. It would strengthen the review to discuss certain challenges such as: Are sensors reliably able to detect under highly strained and deformed? Can sensors differentiate resistance between the strain-induced resistance change and the change in resistance due to hydrogen absorption?

Response: We appreciate the reviewer's insightful comment regarding the challenges faced by stretchable and flexible sensors. To address the challenges, we added a detailed discussion on the challenges of stretchable and flexible hydrogen sensors in Section 3, with a particular focus on their performance under deformation and the differentiation between strain-induced and hydrogen-induced resistance changes.

According to the reviewer's suggestion, we added a new section addressing the major challenges faced by flexible and stretchable hydrogen sensors. This new section is included within Section 3 "Stretchable electronics for hydrogen sensors" under the title "3.3 Challenges in stretchable hydrogen sensors". The section has been structured as follows:

In the new subsection 3.3,

3.3 Challenges in stretchable hydrogen sensors

While flexible and stretchable hydrogen sensor technology holds great promise, sev-eral significant challenges must be overcome for practical applications. This section will examine these key challenges and discuss the approaches researchers are currently taking to address them. The first major challenge is sensor reliability under high strain conditions. Flexible and stretchable sensors, by their nature, can be subjected to various forms of deformation. Won et. al. [108] developed ultrasensitive stretchable conductive fibers using percolated Pd nanoparticle networks, which maintained hydrogen sensing capabilities even when stretched up to 100% strain. Additionally, Kim et. al. [63] developed a high-resolution, fast, and shape-conformable hydrogen sensor platform using polymer nanofiber yarn coupled with nanograined Pd@Pt, which demonstrated stable performance under various bending conditions. The second crucial challenge is differentiating between strain-induced resistance changes and hydrogen-induced resistance changes. This is vital for the accuracy and reliability of stretchable hydrogen sensors. Namgung et. al. [109] created sensors that could distinguish between these two effects by transferring palladium nanosheets onto an elastomeric substrate. They incorporated a reference electrode to compensate for strain effects. Similarly, Lee et. al. [60] developed cracked palladium films on an elastomeric substrate, utilizing unique crack formation and propagation characteristics to differentiate between mechanical deformation and hydrogen exposure effects. Long-term stability and fatigue resistance represent the third major challenge. Lim et. al. [62] investigated the durability of their flexible chemical sensors over thousands of bending cycles, highlighting the importance of maintaining sensor performance under repeated deformation. The fourth challenge relates to temperature effects on flexible and stretchable sensors. Temperature variations can significantly impact sensor performance, particularly in relation to their flexible and stretchable nature.

Comment 3. It seems most of hydrogen sensors that the author described use palladium. I would suggest describing the sensing mechanism of hydrogen first before mentioning the positive and negative response. The current version describes the mechanism in the one of the last part.

Response: We thank for the reviewer’s constructive comment. According to the reviewer's suggestion, we have added the fundamental mechanism of hydrogen sensing before section 2.1.1, providing a foundation for understanding the subsequent discussions on positive and negative responses.

Following your suggestion, we have added the following sentences in the revised manuscript:

In line 223–228 in the section 2, “The fundamental mechanism of hydrogen sensing begins with palladium's unique ability to absorb hydrogen [67]. When exposed to hydrogen, palladium forms palladium hydride (PdHx), which causes a notable expansion in the lattice structure of the Pd film. As hydrogen concentration rises, the Pd film undergoes a phase transition from al-pha-phase PdHx (α-PdHx) to beta-phase PdHx (β-PdHx). This transition leads to a significant change in the electrical resistance of the Pd film.”

Comment 4. Although the mechanism with positive-response was well described, the negative sensing mechanism is not well explained

Response: We thank the reviewer for the careful reading. According to the reviewer’s comment, we included the additional explanation of the negative sensing mechanism in section 2.1.2 in the revised manuscript, providing more detail on how the interaction between hydrogen and sensing materials like palladium can lead to a decrease in resistance.

Following your suggestion, we have added the following sentences in the revised manuscript:

In line 376–403 in the section 2.1.2, “In contrast to the more commonly observed positive resistance changes, some hydrogen sensors exhibit a fascinating phenomenon known as negative resistance. This effect, characterized by a decrease in electrical resistance upon hydrogen exposure, arises from complex interactions between hydrogen and the sensing material at the atomic level [79]. The mechanism of negative resistance changes involves several interrelated process-es. When hydrogen molecules encounter the sensor surface, they typically dissociate into atomic hydrogen. These atoms can then be absorbed into the material's lattice, triggering a cascade of effects. Firstly, absorbed hydrogen atoms often donate electrons to the sensing material, potentially increasing the density of charge carriers and thereby reducing electrical resistance (Electron transfer). Secondly, the incorporation of hydrogen can modify the material's electronic band structure, sometimes creating new conduction pathways or altering the Fermi level in ways that enhance conductivity (Band structure modification). Lastly, hydrogen absorption usually causes lattice expansion, which, while increasing resistance in some materials, can actually enhance electron mobility in others, leading to decreased resistance (Lattice expansion). Compared to their positive resistance counter-parts, negative resistance sensors often offer advantages such as faster response times due to rapid electron transfer processes. The decrease in resistance can also be easier to measure in certain circuit configurations, potentially improving sensitivity. However, the choice between positive and negative resistance sensors ultimately depends on specific application requirements, including the range of hydrogen concentrations to be detected, operating conditions, and compatibility with existing systems.”

Comment 5. The author mostly focused on Pd-based sensors. It would be great if the author rationalize why Pd-based sensors are mostly mentioned in this type of applications. A short and brief discussion on the material’s advantages and why such material-based sensor is favored over others in such applications would improve the review

Response: We thank the reviewer for the careful reading and helpful comment. According to the reviewer’s suggestion, we have added the following sentences to the revised manuscript to explain the advantages of Pd materials and Pd-based sensors:

In line 130–151 in the section 1, “Throughout this review, a significant emphasis on palladium-based sensors may have been noticed. This focus is not arbitrary but stems from palladium's unique proper-ties that make it exceptionally suitable for hydrogen sensing applications, particularly in flexible and stretchable configurations [5]. Palladium's prominence in hydrogen sensing can be attributed to several key characteristics. First and foremost is its remarkable ability to absorb hydrogen – up to 900 times its own volume at room temperature and atmospheric pressure . This exceptional absorption capacity translates into high sensitivity, al-lowing palladium-based sensors to detect even low concentrations of hydrogen. Moreover, palladium exhibits high selectivity towards hydrogen compared to other gases, minimizing false positive readings and enhancing overall sensor reliability [45]. The interaction between palladium and hydrogen is not only highly selective but also rapid and reversible. Palladium quickly absorbs and releases hydrogen, enabling fast response and recovery times in sensing applications [43].

While other materials such as platinum and certain metal oxides are also used in hydrogen sensing, palladium often emerges as the preferred choice for flexible and stretchable sensors due to its malleability and ductility. These properties allow palladium to maintain its sensing capabilities even when subjected to bending, stretching, or other deformations – a crucial requirement for flexible sensor applications. Additionally, palladium can be easily deposited as thin films or nanostructures on various substrates, including flexible polymers, using techniques such as sputtering, electrodeposition, or chemical vapor deposition. This versatility in fabrication makes palladium well-suited for integration into diverse flexible sensor designs.

Reviewer 2 Report

Comments and Suggestions for Authors

This review written by Donggeun Yoo, Kwon-Pil Park and Minsoo P. Kim focuses on the latest advances in the design of flexible and stretchable electronics for hydrogen detection. The review is relevant because everyone understands the role of hydrogen as a clean energy carrier for the entire energy sector of the future. At the same time, it is necessary to take into account the explosive nature of hydrogen, which is why sensors are so important. Modern realities require miniaturization of devices and sensors, which makes them cheaper to use and creates backup systems for increased reliability. The review will be very useful for both students and professionals in this field. The review begins with an introduction to the fundamental principles underlying the operation of flexible and stretchable sensors, highlighting innovative materials and fabrication methods that provide their exceptional mechanical stability and adaptability. After that, the article discusses specific strategies used to integrate these sensors into hydrogen detection systems, discussing the advantages and limitations of various sensor designs, from nanoscale transducers to fully integrated wearable devices. Particular attention is paid to the sensitivity, selectivity, and operational stability of these sensors, as well as their performance in real-world conditions. The authors also discuss the importance of sensor network integration and the development of safety and performance standards. Finally, the review concludes with a perspective look at the potential impact of flexible and stretchable electronics in hydrogen energy systems and safety applications. It can also be noted that the review is well illustrated and well written. The number of typos is minimal.

I have a question. Are there examples, data on sensors that respond differently to para- and ortho-hydrogen? Is there data on different responses to hydrogen and deuterium. If so, it is worth briefly discuss it, that will make the review even more interesting for the reader.

Comments on the Quality of English Language

Minor editing of English language required.

Author Response

This review written by Donggeun Yoo, Kwon-Pil Park and Minsoo P. Kim focuses on the latest advances in the design of flexible and stretchable electronics for hydrogen detection. The review is relevant because everyone understands the role of hydrogen as a clean energy carrier for the entire energy sector of the future. At the same time, it is necessary to take into account the explosive nature of hydrogen, which is why sensors are so important. Modern realities require miniaturization of devices and sensors, which makes them cheaper to use and creates backup systems for increased reliability. The review will be very useful for both students and professionals in this field. The review begins with an introduction to the fundamental principles underlying the operation of flexible and stretchable sensors, highlighting innovative materials and fabrication methods that provide their exceptional mechanical stability and adaptability. After that, the article discusses specific strategies used to integrate these sensors into hydrogen detection systems, discussing the advantages and limitations of various sensor designs, from nanoscale transducers to fully integrated wearable devices. Particular attention is paid to the sensitivity, selectivity, and operational stability of these sensors, as well as their performance in real-world conditions. The authors also discuss the importance of sensor network integration and the development of safety and performance standards. Finally, the review concludes with a perspective look at the potential impact of flexible and stretchable electronics in hydrogen energy systems and safety applications. It can also be noted that the review is well illustrated and well written. The number of typos is minimal.

Response: We would like to thank referees for their encouraging evaluation and positive comments on our manuscript. The typos have been carefully revised.

I have a question. Are there examples, data on sensors that respond differently to para- and ortho-hydrogen? Is there data on different responses to hydrogen and deuterium? If so, it is worth briefly discuss it, that will make the review even more interesting for the reader.

Response: We thank the reviewer for their careful reading and insightful question regarding the potential differentiation between para- and ortho-hydrogen, as well as between hydrogen and deuterium in flexible and stretchable sensors. This is indeed an interesting area that could potentially enhance the scope and depth of our discussion on hydrogen sensing.

While the conversion between para- and ortho-hydrogen is a well-known phenomenon in hydrogen physics and chemistry, as introduced in the review (Frontiers in Chemistry, 2023, 11, 1258035), it's important to note that the flexible and stretchable hydrogen sensors discussed in our review are not specifically designed to differentiate between these nuclear spin isomers. The sensors we've focused on primarily detect the presence and concentration of hydrogen gas, regardless of its nuclear spin state.

Although our review has focused on recent advancements in flexible and stretchable hydrogen sensors, an intriguing area for future development lies in the detection and differentiation of hydrogen isotopes. While the sensors we've discussed primarily detect the presence and concentration of hydrogen gas regardless of its isotopic composition, recent studies have shown promising results in distinguishing between hydrogen isotopes, particularly protium and deuterium. For instance, Matsumoto et al. (Journal of Nuclear Science and Technology, 2002, 39(4), 367-370) developed an electrochemical hydrogen isotope sensor using a proton-conducting electrolyte of CaZr0.90In0.10O3-α. This solid electrolyte-based sensor demonstrated the ability to generate distinct electromotive forces (EMFs) for different hydrogen isotopes. In a different approach, Marcu and Viespe (Sensors, 2017, 17(6), 1417) created a surface acoustic wave (SAW) sensor using zinc oxide (ZnO) nanowires and thin films as sensitive layers, capable of detecting both hydrogen and deuterium. Further advancements in this field were made by Iordache et al. (International Journal of Hydrogen Energy, 2021, 46(19), 11015-11024) and Hu et al. (Sensors and Actuators B: Chemical, 2022, 356, 131344), who independently developed amperometric sensors for hydrogen isotope detection. Both teams utilized Pt nanoparticles confined within carbon nanotubes, showcasing the potential of nanostructured materials in isotope-sensitive sensing.

However, it's important to note that these isotope-sensitive detection methods have not yet been integrated into flexible and stretchable sensor designs. This gap presents an exciting opportunity for future research and development in the field of hydrogen sensing.

Looking ahead, the integration of isotope-sensitive detection mechanisms into flexible and stretchable hydrogen sensors could significantly expand their capabilities and application range. Such advanced sensors could not only detect the presence and concentration of hydrogen but also distinguish between different isotopes. This capability would be particularly valuable in applications related to nuclear energy, fusion research, and isotope tracing in various scientific fields. Therefore, future research should focus on combining the flexibility and stretchability of current sensor designs with the isotope sensitivity demonstrated in recent studies. The development of such advanced sensors would require interdisciplinary collaboration, bringing together expertise in materials science, sensor technology, and isotope chemistry. While challenging, this direction of research has the potential to open new avenues in fields such as nuclear safety, fusion energy research, and environmental monitoring.

While significant progress has been made in the development of flexible and stretchable hydrogen sensors, the integration of isotope-sensitive detection capabilities represents an exciting frontier for future research. By pursuing this direction, a new generation of sensors that combine flexibility, stretchability, and the ability to differentiate between hydrogen isotopes can be created, further expanding the utility and applications of these innovative sensing technologies.

According to the reviewer’s suggestion, we have added the following sentences in the revised manuscript:

In line 844–857 in the section 5, “While our review has focused on the recent advancements in flexible and stretchable hydrogen sensors, an intriguing area for future development lies in the detection and differentiation of hydrogen isotopes. Recent studies have shown promising results in distinguishing between hydrogen isotopes, particularly protium and deuterium, using various sensing technologies [120-124]. However, these isotope-sensitive detection methods have not yet been integrated into flexible and stretchable sensor designs. This gap presents an exciting opportunity for future research and development. The integration of iso-tope-sensitive detection mechanisms into flexible and stretchable hydrogen sensors could significantly expand their capabilities and application range, particularly in fields such as nuclear energy, fusion research, and isotope tracing. The development of such advanced sensors would require interdisciplinary collaboration, bringing together expertise in materials science, sensor technology, and isotope chemistry. While challenging, this direction of research has the potential to open new avenues in fields such as nuclear safety, fusion energy research, and environmental monitoring.”

Reviewer 3 Report

Comments and Suggestions for Authors

The manuscript entitled "Advancements in Flexible and Stretchable Electronics for Hydrogen Sensing: A Comprehensive Review" by Donggeun Yoo et al. provides a thorough overview of the recent progress in flexible and stretchable electronics tailored for hydrogen sensing applications. I am pleased to review this high-quality manuscript, which is not only comprehensive in research but also highly insightful for industrial development. I recommend considering this manuscript for acceptance, although there are some minor issues that need further consideration.

1. The introduction provides a comprehensive background on flexible and stretchable electronics, but it lacks a clear explanation of the unique contributions to the field of hydrogen sensing applications. It is suggested to further emphasize the uniqueness of this review, particularly by highlighting its innovations compared to existing reviews and how it addresses current bottlenecks in sensor technology.

2. Sections 2 and 3, which discuss the selection of sensor materials and their performance, could provide more detail on why certain materials are more suitable for hydrogen sensing than others. It is recommended to include the rationale behind the choice of different materials (e.g., metal oxides, nanomaterials) and to expand the analysis of how these materials perform in real-world applications.

3. While the manuscript discusses the innovations in flexible and stretchable electronics, it lacks concrete examples of applications or discussions on how to overcome current technical challenges. A specific application case could be added to illustrate how the current technological limitations are addressed in practical hydrogen detection scenarios.

4. The conclusion section focuses more on summarizing existing technologies, with a lack of specific discussion on future challenges. It would be beneficial to include a more detailed outlook on future trends in technology development, especially regarding miniaturization of sensors, energy efficiency improvements, and integration with the Internet of Things (IoT).

Author Response

The manuscript entitled "Advancements in Flexible and Stretchable Electronics for Hydrogen Sensing: A Comprehensive Review" by Donggeun Yoo et al. provides a thorough overview of the recent progress in flexible and stretchable electronics tailored for hydrogen sensing applications. I am pleased to review this high-quality manuscript, which is not only comprehensive in research but also highly insightful for industrial development. I recommend considering this manuscript for acceptance, although there are some minor issues that need further consideration.

Response: We would like to thank referees for their encouraging evaluation and constructive comments on our manuscript. In accordance to comments/suggestions from reviewers, careful revision has been made. The revised parts were highlighted in red.

Comment 1. The introduction provides a comprehensive background on flexible and stretchable electronics, but it lacks a clear explanation of the unique contributions to the field of hydrogen sensing applications. It is suggested to further emphasize the uniqueness of this review, particularly by highlighting its innovations compared to existing reviews and how it addresses current bottlenecks in sensor technology.

Response: We thank the reviewer for the careful reading and helpful comment. According to the reviewer’s suggestion, We have revised the introduction in the manuscript to highlight how this review differs from existing ones, specifically by focusing on flexible and stretchable electronics for hydrogen sensing as follows:

In line 98–129 in the section 1, “While several reviews have addressed various aspects of hydrogen sensing technologies [5, 12, 43, 44], this review uniquely focuses on the recent advancements in flexible and stretchable electronics for hydrogen sensing applications. Our work distinguishes it-self by comprehensively examining the innovative materials and fabrication techniques that enable exceptional mechanical resilience and adaptability in these sensors, which is crucial for wearable and on-body applications in dynamic hydrogen mobility environments.

This review uniquely addresses the current challenges and advancements in flexible and stretchable electronics for hydrogen sensing applications, with a particular focus on wearable technologies for hydrogen mobility infrastructure. We comprehensively examine the innovative materials and fabrication techniques that enable exceptional mechanical resilience and adaptability in these sensors, which is crucial for wearable and on-body applications in dynamic hydrogen mobility environments. Our work explores how these advanced sensors overcome limitations of traditional rigid sensors, particularly in applications requiring conformability to complex body geometries or integration into clothing and personal protective equipment. We address current bottlenecks in sensor technology, such as the need for improved sensitivity, faster response times, and better durability un-der mechanical stress - all critical factors in developing reliable wearable hydrogen sensors for mobility applications. Furthermore, we provide an in-depth analysis of the specific strategies employed in integrating these sensors into hydrogen detection systems for mobility infrastructure, discussing the merits and limitations of various sensor configurations including flexible patches, smart textiles, and stretchable bands.

By focusing on these cutting-edge developments in the context of wearable applications for hydrogen mobility, our review aims to bridge the gap between fundamental re-search in materials science and practical, on-body applications in hydrogen sensing technology. These advancements leverage the benefits of flexibility and stretchability to meet the stringent requirements of modern hydrogen energy applications, thereby enhancing hydrogen safety and utility in dynamic environments, from automotive applications to industrial safety monitoring. We believe this comprehensive approach, with its unique focus on wearability and real-world usability in hydrogen mobility scenarios, will provide valuable insights for researchers and engineers working towards the next generation of hydrogen sensors for mobility applications.”

Comment 2. Sections 2 and 3, which discuss the selection of sensor materials and their performance, could provide more detail on why certain materials are more suitable for hydrogen sensing than others. It is recommended to include the rationale behind the choice of different materials (e.g., metal oxides, nanomaterials) and to expand the analysis of how these materials perform in real-world applications.

Response: We thank the reviewer for the valuable feedback. According to the reviewer’s suggestion, we have added the following sentences to each section of the revised manuscript to discuss the rationale behind the choice of different materials and to expand the analysis of sensing performance in real-world applications:

In line 130–151 in the section 1, “Throughout this review, a significant emphasis on palladium-based sensors may have been noticed. This focus is not arbitrary but stems from palladium's unique proper-ties that make it exceptionally suitable for hydrogen sensing applications, particularly in flexible and stretchable configurations [5]. Palladium's prominence in hydrogen sensing can be attributed to several key characteristics. First and foremost is its remarkable ability to absorb hydrogen – up to 900 times its own volume at room temperature and atmospheric pressure . This exceptional absorption capacity translates into high sensitivity, al-lowing palladium-based sensors to detect even low concentrations of hydrogen. Moreover, palladium exhibits high selectivity towards hydrogen compared to other gases, minimiz-ing false positive readings and enhancing overall sensor reliability [45]. The interaction between palladium and hydrogen is not only highly selective but also rapid and reversible. Palladium quickly absorbs and releases hydrogen, enabling fast response and recovery times in sensing applications [43].

While other materials such as platinum and certain metal oxides are also used in hydrogen sensing, palladium often emerges as the preferred choice for flexible and stretchable sensors due to its malleability and ductility. These properties allow palladium to maintain its sensing capabilities even when subjected to bending, stretching, or other deformations – a crucial requirement for flexible sensor applications. Additionally, palladium can be easily deposited as thin films or nanostructures on various substrates, including flexible polymers, using techniques such as sputtering, electrodeposition, or chemical vapor deposition. This versatility in fabrication makes palladium well-suited for integration into diverse flexible sensor designs.”

In line 242–248 in the section 2.1.1, “Palladium and its alloys are frequently used in hydrogen sensors due to their high hy-drogen solubility and fast kinetics [68]. For example, Pd-Au alloys have demonstrated improved stability and faster response times compared to pure Pd [69, 70]. Moreover, metal oxides such as SnO2 and WO3 are chosen for their stability and selectivity [21, 22, 71, 72]. These materials can be fabricated into nanostructures that enhance sensor performance. For instance, WO3 nanowires have shown a response time of less than 20 seconds for 1% hydrogen concentration [71, 72].”

In line 274–278 in the section 2.1.1, “Carbon nanotubes (CNTs) have emerged as a promising material due to their high sur-face-to-volume ratio and excellent electronic properties. For instance, single-walled carbon nanotubes (SWCNTs) decorated with palladium nanoparticles have shown exceptional sensitivity to hydrogen, with detection limits as low as 3 ppm [30].”

In line 689–711 in the section 3.2, “In real-world practical applications, the performance of flexible hydrogen sensors is critical. Studies have shown that these sensors can maintain their performance under varying temperature and humidity conditions [111]. For example, graphene-based flexible hydrogen sensors have demonstrated stable performance in the temperature range of -50°C to 100°C [112], which is suitable for practical applications. Long-term stability is another crucial factor. Extended durability tests have shown that some flexible sensors can maintain their sensitivity for over 1000 hours of continuous operation [113]. This is particularly important for sensors integrated into vehicles or stationary infrastructure for practical applications such as monitoring and safety systems, and fuel cell technology. In terms of response time, flexible sensors based on Pd nanoparticles have achieved response times as low as 1 second for 1% hydrogen concentration, which is faster than many conventional rigid sensors [53, 114].

More specifically, in portable hydrogen detection devices, used by maintenance workers in hydrogen facilities, chemiresistive sensors based on Pd nanoparticles have shown promise [8, 115, 116]. These sensors offer a combination of flexibility, fast response times, and high sensitivity, making them ideal for personal safety equipment. Another emerging application is in smart textiles for hydrogen facility workers. Flexible sensors based on 2D materials have been integrated into fabrics [117, 118], creating wearable safe-ty devices that can alert workers to hydrogen leaks while being comfortable and unobtrusive. For comprehensive monitoring of large hydrogen infrastructure, distributed sensor networks using various flexible sensor types have been proposed [119]. These networks can provide real-time, wide-area hydrogen detection, crucial for the safe operation of hydrogen mobility infrastructure.”

Comment 3. While the manuscript discusses the innovations in flexible and stretchable electronics, it lacks concrete examples of applications or discussions on how to overcome current technical challenges. A specific application case could be added to illustrate how the current technological limitations are addressed in practical hydrogen detection scenarios.

Response: We thank the reviewer for the careful reading and helpful comment. Considering the reviewer’s suggestion, we have added the following sentences to the revised manuscript to discuss the current technical challenges and their specific applications:

In line 689–711 in the section 3.2, “In real-world practical applications, the performance of flexible hydrogen sensors is critical. Studies have shown that these sensors can maintain their performance under varying temperature and humidity conditions [111]. For example, graphene-based flexible hydrogen sensors have demonstrated stable performance in the temperature range of -50°C to 100°C [112], which is suitable for practical applications. Long-term stability is another crucial factor. Extended durability tests have shown that some flexible sensors can maintain their sensitivity for over 1000 hours of continuous operation [113]. This is particularly important for sensors integrated into vehicles or stationary infrastructure for practical applications such as monitoring and safety systems, and fuel cell technology. In terms of response time, flexible sensors based on Pd nanoparticles have achieved response times as low as 1 second for 1% hydrogen concentration, which is faster than many conventional rigid sensors [53, 114].

More specifically, in portable hydrogen detection devices, used by maintenance workers in hydrogen facilities, chemiresistive sensors based on Pd nanoparticles have shown promise [8, 115, 116]. These sensors offer a combination of flexibility, fast response times, and high sensitivity, making them ideal for personal safety equipment. Another emerging application is in smart textiles for hydrogen facility workers. Flexible sensors based on 2D materials have been integrated into fabrics [117, 118], creating wearable safety devices that can alert workers to hydrogen leaks while being comfortable and unobtrusive. For comprehensive monitoring of large hydrogen infrastructure, distributed sensor networks using various flexible sensor types have been proposed [119]. These networks can provide real-time, wide-area hydrogen detection, crucial for the safe operation of hydrogen mobility infrastructure.”

In the new subsection 3.3,

3.3 Challenges in stretchable hydrogen sensors

While flexible and stretchable hydrogen sensor technology holds great promise, sev-eral significant challenges must be overcome for practical applications. This section will examine these key challenges and discuss the approaches researchers are currently taking to address them. The first major challenge is sensor reliability under high strain conditions. Flexible and stretchable sensors, by their nature, can be subjected to various forms of deformation. Won et. al. [108] developed ultrasensitive stretchable conductive fibers using percolated Pd nanoparticle networks, which maintained hydrogen sensing capabilities even when stretched up to 100% strain. Additionally, Kim et. al. [63] developed a high-resolution, fast, and shape-conformable hydrogen sensor platform using polymer nanofiber yarn coupled with nanograined Pd@Pt, which demonstrated stable performance under various bending conditions. The second crucial challenge is differentiating between strain-induced resistance changes and hydrogen-induced resistance changes. This is vital for the accuracy and reliability of stretchable hydrogen sensors. Namgung et. al. [109] created sensors that could distinguish between these two effects by transferring palladium nanosheets onto an elastomeric substrate. They incorporated a reference electrode to compensate for strain effects. Similarly, Lee et. al. [60] developed cracked palladium films on an elastomeric substrate, utilizing unique crack formation and propagation characteristics to differentiate between mechanical deformation and hydrogen exposure effects. Long-term stability and fatigue resistance represent the third major challenge. Lim et. al. [62] investigated the durability of their flexible chemical sensors over thousands of bending cycles, highlighting the importance of maintaining sensor performance under repeated deformation. The fourth challenge relates to temperature effects on flexible and stretchable sensors. Temperature variations can significantly impact sensor performance, particularly in relation to their flexible and stretchable nature.

Comment 4. The conclusion section focuses more on summarizing existing technologies, with a lack of specific discussion on future challenges. It would be beneficial to include a more detailed outlook on future trends in technology development, especially regarding miniaturization of sensors, energy efficiency improvements, and integration with the Internet of Things (IoT).

Response: We thank the reviewer for the constructive comment. According to the reviewer’s suggestion to expand the conclusion with a more detailed outlook on future trends, we have added the following sentences in the revised manuscript:

In line 825–843 in the section 5, “Looking ahead, several key areas of development are poised to significantly advance flexible and stretchable hydrogen sensors for next generation applications. Advancements in nanotechnology and microfabrication techniques are driving the miniaturization of sensors, potentially to the nanoscale, enabling more comprehensive and granular hydro-gen monitoring in vehicles and infrastructure. Concurrently, efforts to improve energy efficiency may lead to the development of self-powered sensors leveraging energy harvest-ing technologies, crucial for wearable applications and remote deployments. The integra-tion of these sensors with the IoT will allow for real-time, wireless monitoring across en-tire mobility ecosystems, while the development of multi-functional sensors could com-bine hydrogen detection with other critical measurements such as temperature or pressure. Enhancing the durability and reliability of these sensors under real-world conditions re-mains a key challenge, with research focusing on new protective coatings and encapsulation techniques. To support widespread adoption, advancements in large-scale manufacturing techniques, such as roll-to-roll processing and 3D printing, will be vital. Innovative approaches, including bio-inspired designs, might lead to self-healing sensors capable of recovering from mechanical damage, significantly extending their lifespan in harsh environments. Finally, as the field matures, the development of international standards and regulatory frameworks will be crucial to ensure reliability and interoperability across different hydrogen mobility platforms and infrastructure.”
